# Simulating ozone dry deposition at a boreal forest with a multi-layer canopy deposition model

Putian Zhou[1], Laurens Ganzeveld[2], Üllar Rannik[1], Luxi Zhou[1, *], Rosa Gierens[1, **], Ditte Taipale[3,4], Ivan Mammarella[1], and Michael Boy[1]

[1]University of Helsinki, Department of Physics, P.O. Box 64, FI-00014, University of Helsinki, Finland
[2]Meteorology and Air Quality (MAQ), Department of Environmental Sciences, Wageningen University and Research Centre, Wageningen, Netherlands
[3]University of Helsinki, Department of Forest Sciences, P.O. Box 27, FI-00014, University of Helsinki, Finland
[4]Estonian University of Life Sciences, Department of Plant Physiology, Kreutzwaldi 1, EE-51014, Estonia
[*]now at U.S. Environmental Protection Agency, Research Triangle Park, NC, USA
[**]now at Institute for Geophysics and Meteorology, University of Cologne, Germany
*Correspondence to:* Zhou Putian (putian.zhou@helsinki.fi)

**Abstract.** A multi-layer ozone ($O_3$) dry deposition model has been implemented into SOSAA (a model to Simulate the concentrations of Organic vapours, Sulphuric Acid and Aerosols) to improve the representation of $O_3$ concentration and flux within and above the forest canopy in the planetary boundary layer. We aim to predict the $O_3$ uptake by a boreal forest canopy under varying environmental conditions and analyse the influence of different factors on total $O_3$ uptake by the canopy as well as the vertical distribution of deposition sinks inside the canopy. The newly implemented dry deposition model was validated by an extensive comparison of simulated and observed $O_3$ turbulent fluxes and concentration profiles within and above the boreal forest canopy at SMEAR II (the Station to Measure Ecosystem-Atmosphere Relation II) in Hyytiälä, Finland, in August, 2010.

In this model, the fraction of wet surface on vegetation leaves was parametrised according to the ambient relative humidity (RH). Model results showed that when RH was larger than 70% the $O_3$ uptake onto wet skin contributed $\sim 51\%$ to the total deposition during nighttime and $\sim 19\%$ during daytime. The overall contribution of soil uptake was estimated about 36%. The contribution of sub-canopy deposition below 4.2 m was modelled to be $\sim 38\%$ of the total $O_3$ deposition during daytime which was similar to the contribution reported in previous studies. The chemical contribution to $O_3$ removal was evaluated directly in the model simulations. According to the simulated averaged diurnal cycle the net chemical production of $O_3$ compensated up to $\sim 4\%$ of dry deposition loss from about 06:00 to 15:00. During nighttime, the net chemical loss of $O_3$ further enhanced removal by dry deposition by a maximum $\sim 9\%$. Thus the results indicated an overall relatively small contribution of airborne chemical processes to $O_3$ removal at this site.

## 1  Introduction

Tropospheric ozone ($O_3$) is an important oxidant of many reactive species such as biogenic volatile organic compounds (BVOCs) emitted from the forest canopy (Bäck et al., 2012; Smolander et al., 2014). It also plays a significant role in the regulation of the atmospheric oxidation capacity first of all by being one of the primary sources of the hydroxyl radical (OH)

which is the most critical oxidant in the air (Mogensen et al., 2015). $O_3$ also initiates the formation of Criegee intermediate (CI) radicals which are crucial in tropospheric oxidation (Boy et al., 2013). As an air pollutant, $O_3$ can cause damage to human health (Kampa and Castanas, 2008) and affect ecosystem functioning via its various toxic impacts (Felzer et al., 2007). $O_3$ can also alter the global radiative forcing as an important greenhouse gas (Stocker et al., 2013, chap. 2). Hence it is impor-
tant to understand the $O_3$ budget including its sources and sinks at local or site scale in order to understand the global scale implications.

$O_3$ is produced via photochemical reactions in the presence of precursor gases, e.g., volatile organic compounds (VOCs), CO (carbon oxide), OH and $NO_x$ (nitric oxide and nitrogen dioxide) or transported downward from stratosphere, and is removed mainly near the Earth's surface. For vegetated surfaces a large part of the removal processes occur via stomatal uptake on leaf surface and non-stomatal uptake on plant canopies and soil surface (Wesely, 1989; Ganzeveld and Lelieveld, 1995; Altimir et al., 2006; Rannik et al., 2012; Launiainen et al., 2013), as well as depletion by chemical reactions (Kurpius and Goldstein, 2003; Wolfe et al., 2011). In this study we focus on the $O_3$ removal and production processes within and immediately above the canopy, more particularly on the $O_3$ uptake by boreal forest which covers 33% of global forest land (Ruckstuhl et al., 2008).

For vegetation, the uptake of $O_3$ depends on the turbulence intensity above and within the canopy, the diffusive transfer in the quasi-laminar boundary layer over the leaf surface, the biological properties of the plants, surface wetness condition, and soil type (Ganzeveld and Lelieveld, 1995). Among them the effect of canopy wetness on $O_3$ deposition has attracted a lot of attention in previous studies (e.g., Massman, 2004; Altimir et al., 2006). For different vegetation types and under different environmental conditions the surface wetness can enhance or reduce $O_3$ deposition (Massman, 2004). For a boreal forest, a number of studies have revealed an enhancement of the $O_3$ uptake under dew or high humidity conditions. For example, Lamaud et al. (2002) reported that dew on canopy surface significantly increased the $O_3$ uptake at night and in the morning over a pine stand. Altimir et al. (2006) also found that the condensed moisture on the surfaces enhanced the non-stomatal $O_3$ uptake in a Scots pine forest when ambient relative humidity (RH) was over 60 - 70%. Similarly to Altimir et al. (2006), Rannik et al. (2012) revealed a strong sensitivity of the nighttime $O_3$ uptake to RH. The enhancement of $O_3$ uptake on wet leaf surface was explained by previous studies with both the micro structure of the leaf surface and the hydrophilic compounds existing on the leaf surface which are able to facilitate the formation of the water films or clusters, although the foliage surface itself is hydrophobic (Altimir et al., 2006). As a result, the different dissolved compounds like organics in the solution formed on leaf surface could react with $O_3$ and thus enhance the $O_3$ uptake (Altimir et al., 2006).

In addition, the boreal forest emits a large portion of BVOCs (Rinne et al., 2009) which are considered to play a significant role in non-stomatal removal of $O_3$ by oxidation (Kurpius and Goldstein, 2003; Goldstein et al., 2004; Wolfe et al., 2011). For example, Fares et al. (2010) found the correlation between the oxidation products of monoterpenes and $O_3$ non-stomatal flux at a ponderosa pine stand in California, US, indicating that the gas-phase reactions of $O_3$ with BVOCs were mostly responsible for $O_3$ non-stomatal loss. In a model study, Wolfe et al. (2011) suggested that the non-stomatal $O_3$ uptake at the same Californian site could be explained by considering the role of $O_3$ destruction with the presence of very reactive BVOCs. Consequently, further analysis of the role of non-stomatal removal of $O_3$ also strongly depends on the improvement of BVOCs measurement. However, the influence of this gas-phase chemical removal process may vary among different sites. A study by Rannik et al.

(2012), who conducted a detailed analysis of a long-term $O_3$ deposition flux measurement at the same site as in this study (SMEAR II, a boreal forest station in Hyytiälä, Finland), indicated that, at the currently known strength of BVOC emissions, the air chemistry of BVOCs was not likely an important $O_3$ sink term at this site.

In recent two decades, several numerical models have been developed to study and simulate $O_3$ dry deposition processes under different climatic and environmental conditions. Many of them have implemented the big-leaf framework following the Wesely (1989) approach which can be coupled to regional or global models to estimate the $O_3$ deposition flux in large scales (e.g., Hardacre et al., 2015). However, the "big-leaf" approach does not consider explicitly the role of in-canopy interactions between biogenic emissions, chemistry, turbulence and deposition. Therefore, more detailed multi-layer models including the role of these in-canopy interactions have been developed and applied to analyse in-canopy deposition-related mechanisms (e.g., Ganzeveld et al., 2002b; Rannik et al., 2012; Launiainen et al., 2013). These multi-layer canopy exchange models have also been coupled to large scale models, e.g., a global chemistry-climate model system (Ganzeveld et al., 2002a), or have been implemented in column models with detailed vertically separated layers (e.g., Wolfe and Thornton, 2011).

In this study a multi-layer $O_3$ dry deposition model was implemented into the 1-dimension (1D) chemical transport model SOSAA (a model to Simulate the concentrations of Organic vapours, Sulphuric Acid and Aerosols). This deposition model was based on the dry deposition representation originally described in Ganzeveld and Lelieveld (1995) and Ganzeveld et al. (1998) and implemented in the Multi-Layer Canopy CHemistry Exchange Model (MLC-CHEM, Ganzeveld et al. (2002b)). This canopy exchange system in MLC-CHEM was already applied in a single column model on the analysis of site-scale exchange processes (Ganzeveld et al., 2002b; Seok et al., 2013), as well as in a global chemistry-climate model system on the analysis of atmosphere-biosphere exchange processes (Ganzeveld et al., 2002a, 2010).

Furthermore, the long-term continuous measurements and extensive campaigns at SMEAR II have provided a vast amount of data with complementary information on micrometeorology as well as $O_3$ fluxes and concentrations, which are highly appropriate for validating the new model and investigating more detailed processes. We selected a featured month August 2010 for such an extensive evaluation of the model because this month was characterised by exceptional hot and dry conditions in the first two weeks, which possibly represented a future climate at this site (Williams et al., 2011), then followed by two cooler weeks. This study is a starting point of investigating gas dry deposition processes by using SOSAA. We aim to evaluate not only quantitatively $O_3$ fluxes and concentration profiles but also the role of individual deposition processes at this site. This is a prerequisite for a further analysis of BVOCs deposition and chemistry in the follow-up research.

In the following section, a detailed description of the measurement and model will be shown. The comparisons between simulated and observed meteorological quantities, $O_3$ fluxes above the canopy and $O_3$ concentration profiles are described in section 3, as well as the discussion about $O_3$ flux profiles and the impact of air chemistry. Finally, a summary is given in section 4.

## 2 Methods

### 2.1 Site

All the measurement data used in this study were from SMEAR II (the Station to Measure Ecosystem-Atmosphere Relation II) located in Hyytiälä, Finland (61°51'N, 24°17'E, 181 m above the sea level) (Hari and Kulmala, 2005). The boreal coniferous

forest is relatively homogeneous around the station in all the directions within 200 m, 75% covered by Scots pine (*Pinus sylvestris*) and the rest covered by Norway Spruces (*Picea abies*) and deciduous trees (Bäck et al., 2012). The understory vegetation mainly consists of lingonberry (*Vaccinium vitis-idaea*) and blueberry (*Vaccinium myrtillus*) with a mean height of 0.2 - 0.3 m. The forest floor is covered by dense mosses, mostly *Dicranum polysetum*, *Hylocomium splendens* and *Pleurozium schreberi*. Underneath is a 5 cm layer of humus in soil (Kolari et al., 2006; Kulmala et al., 2008). In 2010, the tree height

reaches around 18 m. The all-sided leaf area index (LAI) is about 7.5 $m^2 \ m^{-2}$, including $\sim 6.0 \ m^2 \ m^{-2}$ overstory vegetation, $\sim 0.5 \ m^2 \ m^{-2}$ understory vegetation and $\sim 1 \ m^2 \ m^{-2}$ moss layer (Launiainen et al., 2013). The vertical profiles of LAI and leaf area density (LAD) are shown in Fig. 1.

### 2.2 Measurements

The measurement data at SMEAR II are currently publicly available in the data server maintained by AVAA open data pub-

15 lishing platform (http://avaa.tdata.fi/web/smart/smear), which was originally introduced in Junninen et al. (2009). A part of observed quantities used in this study are available at 4.2 m, 8.4 m, 16.8 m, 33.6 m, 50.4 m and 67.2 m above the ground level, including air temperature (measured by Pt100 sensor), air water content (Li-Cor LI-840 infrared light absorption analyser) and $O_3$ concentration (TEI 49C ultraviolet light absorption analyser). Other observed quantities include the photosynthetically active radiation (PAR, 400–700 nm) (Li-Cor Li-190SZ quantum sensor) measured at 18 m, PAR (array of 4 Li-Cor Li-190SZ

sensors) measured at 0.6 m, net radiation (Reeman MB-1 net radiometer) at 67 m, $O_3$ flux (Gill Solent HS 1199 sonic anemometer & Unisearch Associates LOZ-3 gas analyzer) at 23 m, friction velocity (Gill Solent 1012R anemometer/themometer) at 23 m, sensible and latent heat fluxes (H and LE) (Gill Solent 1012R and Li-Cor LI-6262 gas analyzer) at 23 m, and soil heat flux (Hukseflux HFP01 heat flux sensors).

In this study the measured $O_3$ fluxes were calculated over 30 min averaging period using the EddyUH software (Mammarella

et al., 2016) and according to standard methodology (for more details see Rannik et al., 2012). Other variables were also half-hour averaged to fit the model time step for both input and output. The air temperature ($T$), RH and $O_3$ concentration were linearly interpolated using the observations collected at a height of 16.8 m and 33.6 m to arrive at the estimated parameter values at 23 m to allow a direct comparison of the model results with the measurements or being used as input for the model. The missing observed data points of $T$, RH and $O_3$ were gap-filled with the method described in Gierens et al. (2014).

The measured $O_3$ fluxes were filtered based on the fact that previous studies showed that the measured fluxes had large errors under very low turbulence (Rannik et al., 2006). The threshold of such low turbulence condition was usually set according to the measured friction velocity on top of the canopy in the range of 0.1 m $s^{-1}$ to 0.25 m $s^{-1}$ (Altimir et al., 2006; Rannik et al., 2012; Launiainen et al., 2013). Here the observed $O_3$ fluxes were excluded when $u_* \leqslant 0.2$ m $s^{-1}$ which was proposed

by Rannik et al. (2012). Secondly, the $O_3$ flux measurements were filtered out when precipitation occurred within preceding 1 hour. Previous studies used a more strict criteria for such a filter that the preceding 12 hours should keep dry to ensure dry canopy conditions (Altimir et al., 2006; Launiainen et al., 2013). However, in this study the fraction of wet canopy skin was taken into account and consequently we applied the filtering criteria of 1 hour. Overall, 60% of $O_3$ flux data were available compared to 87% prior to filtering.

Here we should notice that the fluxes determined by the eddy-covariance (EC) technique were affected by the stochastic nature of turbulence, revealing as the random uncertainty of 30 min average fluxes. For the EC measurement the random uncertainty was typically in the order of ten to a few tens of percent. For the $O_3$ turbulent flux measurement at the same site Keronen et al. (2003) presented the random error statistics, defined as one standard deviation of the random uncertainty of turbulent flux, ranging from about 10 to 40%.

## 2.3 Classification of time period

Previous studies showed that in pine forest RH could enhance non-stomatal $O_3$ uptake (Lamaud et al., 2002; Altimir et al., 2006; Rannik et al., 2012), especially during nighttime (Rannik et al., 2012). Hence in order to further analyse the impact of RH, the data were separated into different groups according to daytime (D) and nighttime (N) as well as RH measured inside the canopy, representing the daytime with high humidity condition (DH), daytime with low humidity condition (DL), nighttime with high humidity condition (NH) and nighttime with low humidity condition (NL). The data points were considered as daytime when the sun elevation angle was larger than $10°$ and as nighttime when the sun elevation angle was smaller than $0°$. The RH threshold value was set to 70% referring to previous studies (Altimir et al., 2006; Rannik et al., 2012), so a period is in high humidity condition when all the measured RH values inside the canopy are higher than 70%, similarly a period is in low humidity condition when all the measured RH values inside the canopy are lower than 70%. For $O_3$ flux, "ALL" was used to represent the time period with all available data after filtering described in section 2.2.

## 2.4 Model description

### 2.4.1 SOSAA

SOSAA is a 1D chemical transport model which couples different modules to simulate the emissions of BVOCs, chemical reactions of organic and inorganic compounds in the air, transportation of trace gases and aerosol particles, as well as the aerosol processes within and above the canopy in the planetary boundary layer. It was first introduced as SOSA by Boy et al. (2011) based on the 1D version of SCADIS (SCAlar DIStribution, Sogachev et al., 2002). After that an aerosol module based on UHMA (University of Helsinki Multicomponent Aerosol model, Korhonen et al., 2004) was implemented by Zhou et al. (2014) resulting in its name being changed to SOSAA. The current version of SOSAA includes five modules. The meteorology module is based on SCADIS. Emissions of BVOCs from the canopy are calculated by the Model of Emissions of Gases and Aerosols from Nature (MEGAN, Guenther et al., 2006). The Master Chemical Mechanism version 3.2 (MCMv3.2) (http://mcm.leeds.ac.uk/MCM) has been implemented to provide chemistry information. The nucleation, condensation, coagulation

and deposition of aerosol particles are described by UHMA. In this study a gaseous compound dry deposition module has been implemented into SOSAA. SOSAA has already been applied and verified in several studies (e.g., Kurtén et al., 2011; Mogensen et al., 2011; Boy et al., 2013; Mogensen et al., 2015; Bäck et al., 2012; Smolander et al., 2014; Zhou et al., 2015).

In SOSAA, the horizontal wind velocity ($u$ and $v$), $T$, specific humidity ($q_v$), turbulent kinetic energy (TKE) and the specific dissipation of TKE ($\omega$) are computed every time step (10 s) by prognostic equations. In order to represent the local to synoptic scale effects, $u$, $v$, $T$ and $q_v$ near and within the canopy are nudged to local measurement data at SMEAR II station with a nudging factor of 0.01. A TKE-$\omega$ parametrisation scheme is used to calculate the turbulent diffusion coefficient ($K_t$) (Sogachev, 2009),

$$K_t = C_\mu \frac{\text{TKE}}{\omega} \tag{1}$$

$$\omega = \frac{\varepsilon}{\text{TKE}} \tag{2}$$

where $\varepsilon$ is the dissipation rate of TKE and $C_\mu$ (0.0436) is a closure constant. Hence the turbulent flux of a quantity $X$ ($F_{t,X}$) can be computed as

$$F_{t,X} = -K_t \frac{\partial X}{\partial z} \tag{3}$$

where upward fluxes are positive and vice versa. Specifically, the sensible heat flux (H) and latent heat flux (LE) at each model layer are computed as

$$\text{H} = -C_{p,air} \rho_{air} K_h \left( \frac{\partial T}{\partial z} + \gamma_d \right) \tag{4}$$

$$\text{LE} = -L_v K_h \frac{\partial q_v}{\partial z} \tag{5}$$

where $C_{p,air}$ (1009.0 J kg$^{-1}$ K$^{-1}$) is the specific heat capacity at constant pressure. $\rho_{air}$ (1.205 kg m$^{-3}$) is the air density which is a constant in the model. $\gamma_d$ (0.0098 K m$^{-1}$) is the lapse rate of dry air. $L_v$ ($2.256 \times 10^6$ J kg$^{-1}$) is the latent heat of vaporisation for water. $K_h$ is the turbulent eddy diffusivity for heat fluxes, which is derived from $K_t$ according to the atmospheric stability.

The upper boundary values of $u$, $v$, $T$ and $q_v$ are constrained by the ERA-Interim reanalysis dataset provided by the European Centre for Medium-Range Weather Forecasts (ECMWF, Dee et al., 2011). Above the canopy, the incoming direct and diffuse global radiations measured at SMEAR II station, and the long wave radiation obtained from the ERA-Interim dataset are read in to improve the energy balance closure. Then the reflection, absorption, penetration and emission of three bands of radiation (long-wave, near-infrared and PAR) at each layer inside the canopy are explicitly computed according to the radiation scheme proposed by Sogachev et al. (2002). At the lower boundary, the measured soil heat flux at SMEAR II is used to further improve the representation of surface energy balance. All the input data are interpolated to match the model time for each time step. With the input data, the mass and energy exchange between atmosphere and plant cover (including the soil underneath) and the radiation attenuation inside the canopy are optimal to simulate the micrometeorological drivers of O$_3$ deposition at this site.

In current SOSAA, a modified version of MEGAN has been used to simulate the emissions of BVOCs from the trees. The emissions of some important BVOCs are included, e.g., monoterpenes ($\alpha$-pinene, $\beta$-pinene, $\Delta^3$-carene, limonene, cineol and

other minor monoterpenes (OMT)), sesquiterpenes (farnesene, $\beta$-caryophyllene and other minor sesquiterpenes (OSQ)) and 2-methyl-3-buten-2-ol (MBO). The chemistry mechanism is from MCMv3.2 including necessary inorganic reactions and the full MCM oxidation paths for methane ($CH_4$), isoprene, MBO, $\alpha$-pinene, $\beta$-pinene, limonene and $\beta$-caryophyllene. We have also included the first-order oxidation reactions with OH, $O_3$, $NO_3$ for cineole, $\Delta^3$-carene, OMT, farnesene and OSQ. The
related chemical reactions of stabilised Criegee intermediates (sCIs) with updated reaction rates from Boy et al. (2013) are also taken into account in current simulations. For more details about emissions and chemistry we refer to Mogensen et al. (2015).

### 2.4.2  Multi-layer $O_3$ dry deposition model

A gas dry deposition model has been implemented into SOSAA to investigate the influence of the dry deposition processes on the atmosphere-biosphere gas exchange and in-canopy gas concentrations. In this study we focus on the $O_3$ dry deposition
since it is the basis of calculating the uptake of other trace gases, including BVOCs (Wesely, 1989). In this multi-layer dry deposition model the $O_3$ deposition flux is calculated at each layer as

$$F_i = -[O_3]_i \cdot V_{d,i} \qquad (i = 1, \ldots, N) \tag{6}$$

where $F$ is the $O_3$ deposition flux ($\mu$g m$^{-2}$ s$^{-1}$), $[O_3]$ is the $O_3$ concentration ($\mu$g m$^{-3}$), $V_d$ is the layer-specific conductance (m s$^{-1}$). The subscript $i$ represents layer index. Layer 1 is the bottom layer including the soil surface and the understory
vegetation where the moss layer is considered as part of the soil surface for simplicity. The overstory layers 2 to $N$ include only vegetation surface, where $N$ is the layer index at the canopy top.

$V_d$ is calculated for bottom layer (layer 1) and overstory layers (layers 2 to $N$) differently. In addition, the deposition onto dry and wet parts of the leaf surface is considered separately. In overstory layers, only the deposition onto leaves is taken into account, while in the bottom layer the additional pathway of deposition onto the soil surface exists. Thus

$$V_{d,i} = \text{LAI}_i V_{dveg,i} + \delta_{i1} V_{dsoil} \tag{7}$$

$$V_{dveg,i} = \frac{1}{r_{veg,i}} \tag{8}$$

$$V_{dsoil} = \frac{1}{r_{ac} + r_{bs} + r_{soil}}. \tag{9}$$

Here $\text{LAI}_i$ is the all-sided leaf area index for each layer (m$^2$ m$^{-2}$). The Kronecker delta $\delta_{i1}$ ($\delta_{i1} = 1$ when $i = 1$; $\delta_{i1} = 0$ when $i \neq 1$) is introduced here to simplify the formula. $V_{dveg,i}$ is the layer-specific leaf surface conductance and $V_{dsoil}$ is the soil
conductance.

$r_{veg}$ is the leaf surface resistance which represents how $O_3$ finally deposits onto different parts of leaf surface (Fig. 1). It can be calculated at each layer for needle leaves as

$$r_{veg} = r_b + \frac{1}{1/(r_{stm} + r_{mes}) + (1 - f_{wet})/r_{cut} + f_{wet}/r_{ws}}. \tag{10}$$

While for broad leaves, $O_3$ can deposit on a side without stomata or a side with stomata, hence $r_{veg}$ is computed in a different way as

$$r_{veg} = 2 \left/ \left( \frac{1}{r_{veg1}} + \frac{1}{r_{veg2}} \right) \right. \tag{11}$$

$$r_{veg1} = r_b + \frac{1}{(1 - f_{wet})/r_{cut} + f_{wet}/r_{ws}} \tag{12}$$

$$r_{veg2} = r_b + \frac{1}{1/(r_{stm} + r_{mes}) + (1 - f_{wet})/r_{cut} + f_{wet}/r_{ws}}. \tag{13}$$

Here $r_b$ is the quasi-laminar boundary layer resistance over the leaf surface, which depends on molecular diffusivity and horizontal wind speed (Meyers, 1987), and $r_{stm}$ is the stomatal resistance which is derived from the stomatal resistance for water vapour ($r_{stm,H_2O}$) by using a factor of the molecular diffusivity ratio,

$$r_{stm} = \frac{D_{H_2O}}{D_{O_3}} r_{stm,H_2O}. \tag{14}$$

Here $D_{H_2O}$ and $D_{O_3}$ are the molecular diffusivities of water vapour and $O_3$, respectively. $r_{stm,H_2O}$ is computed by SCADIS module in SOSAA and also used to calculate latent heat flux and thus the energy balance (Sogachev et al., 2002). $r_{mes}$ is the mesophyllic resistance which can be ignored for $O_3$ (0 s m$^{-1}$). $r_{cut}$ ($10^5$ s m$^{-1}$) is the cuticle resistance and $r_{ws}$ (2000 s m$^{-1}$) represents the uptake on leaf wet skin. Their values are taken from Ganzeveld and Lelieveld (1995). Canopy wetness is represented by the fraction of wet skin $f_{wet}$ which is determined by RH (Lammel, 1999; Wu et al., 2003),

$$f_{wet} = \begin{cases} 1 & 0.9 \leqslant \text{RH} \\ \frac{\text{RH} - 0.7}{0.2} & 0.7 \leqslant \text{RH} < 0.9 \\ 0 & \text{RH} < 0.7 \end{cases}. \tag{15}$$

The threshold 70% is suggested by Altimir et al. (2006).

$r_{ac}$ is the resistance representing the turbulent transport from the reference height of the understory vegetation to the soil surface. Since the gas transport is explicitly calculated in SOSAA and the bottom layer height is only $\sim$ 0.3 m, the turbulence resistance between vegetation and ground is expected not to be an important factor for soil deposition, and consequently we have set $r_{ac}$ to zero. $r_{bs}$ is the soil boundary layer resistance which is calculated as (Nemitz et al., 2000; Launiainen et al., 2013)

$$r_{bs} = \frac{\text{Sc} - \ln(\delta_0/z_*)}{\kappa u_{*g}}. \tag{16}$$

Here Sc (1.07) is the Schmidt number for $O_3$. $\kappa$ is the von Kármán constant (0.41). $\delta_0 = D_{O_3}/(\kappa u_{*g})$ is the height above ground where the molecular diffusivity is equal to turbulent eddy diffusivity. $z_*$ (0.1 m) is the height under which the logarithmic wind profile is assumed. $u_{*g}$ is the friction velocity near the ground. $r_{soil}$ is the soil resistance, 400 s m$^{-1}$ is used here according to Ganzeveld and Lelieveld (1995). A sensitivity analysis for $r_{soil}$ will be shown in section 2.6. The diagram of the resistance analogy parametrisation method described above is shown in Fig. 1. All the symbols are also explained and listed in Table 4.

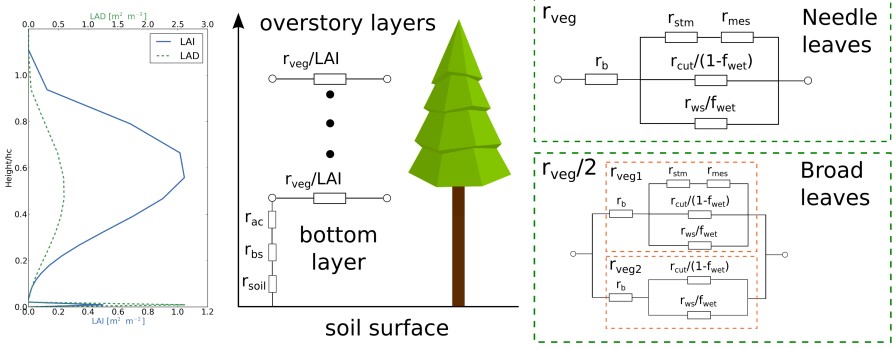

**Figure 1.** Vertical profiles of all-sided LAI (leaf area index) and LAD (leaf area density), as well as the diagram of resistance analogy method used in the $O_3$ dry deposition model. The overstory layers and the bottom layer are considered separately. The bottom layer includes the broad-leaved understory vegetation and soil surface. $r_{ac}$ is the resistance representing the turbulent transport from the reference height of the understory vegetation to the soil surface. $r_{bs}$ is the soil boundary layer resistance. $r_{soil}$ is the soil resistance. $r_b$ is the quasi-laminar boundary layer resistance above the leaf surface. $r_{veg}$ represents the resistance to vegetation leaves, which is plotted on the right-hand side in details. For broad leaves, the resistance to the side with ($r_{veg1}$) or without ($r_{veg2}$) stomata is computed separately. $r_{stm}$ is the stomatal resistance and $r_{mes}$ is the mesophyllic resistance. $r_{cut}$ is the cuticle resistance, $r_{ws}$ is the resistance to wet skin. $f_{wet}$ is the wet skin fraction. All the variables are defined for each layer. Note that here LAI is the all-sided leaf area index for each layer. The symbols are also explained in the text and Table 4.

In the model the evolution of $O_3$ concentration is calculated for each layer by the prognostic equation

$$\frac{\partial [O_3]}{\partial t} = \frac{\partial}{\partial z}\left(K_t \frac{\partial [O_3]}{\partial z}\right) - (V_{dveg}\text{LAD} + V_{dsoil}A_s)[O_3] + Q_{chem} \tag{17}$$

where the first term on the right-hand side represents the vertical mixing of $O_3$. The second term is the sink by dry deposition which is non-zero only inside the canopy. The last one is chemistry production and loss of $O_3$ for each model layer. $A_s$ (m$^2$ m$^{-3}$) is the soil area index which is the ratio between soil area and the model grid volume, hence it is non-zero only at the bottom layer which includes the soil surface. All the other chemical compounds are also computed following this prognostic equation. According to Eq. 3 the $O_3$ turbulent flux $F_t$ in the model can be obtained as

$$F_t = -K_t \frac{\partial [O_3]}{\partial z}. \tag{18}$$

## 2.5 Model setup

In this study the newly implemented $O_3$ dry deposition module was applied to simulate the time period from Aug 1st to Aug 31st 2010 (Julian day 213 to 243). The model column domain was set from 0 m at ground surface up to 3000 m with 51 layers logarithmically configured, including the whole planetary boundary layer and part of the free atmosphere on top of it. We also constrained the model with the site-specific vegetation cover properties as presented before in section 2.1. The overstory layers only included needle-leaved part of Scots pine trees above $\sim 0.3$ m. Below that there was the understory vegetation

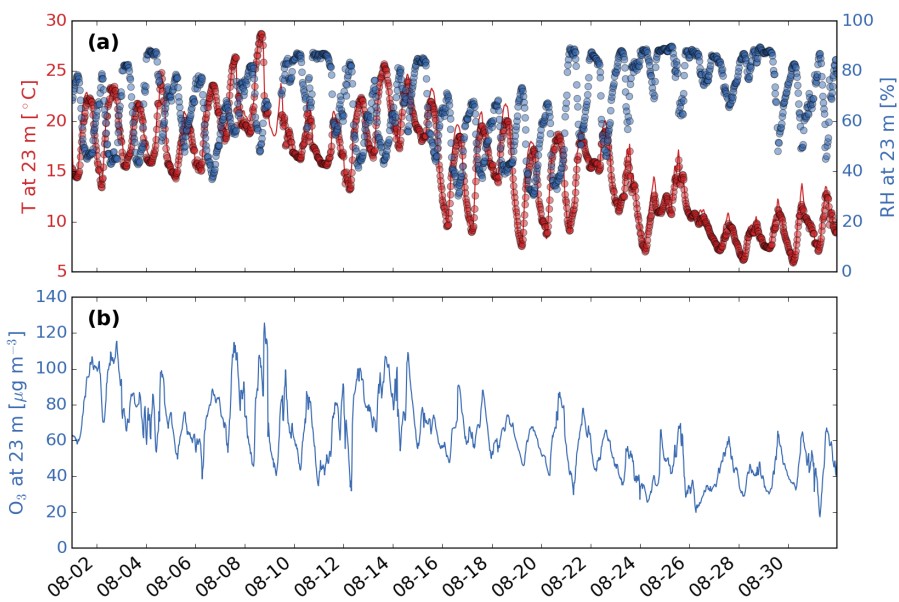

**Figure 2.** (a) Modelled (solid line) and measured (dots) time series of air temperature (T, red) and the measured ambient relative humidity (RH, blue) at 23 m above the ground. (b) Measured $O_3$ concentration (blue) at 23 m above the ground. The time period is August, 2010.

and ground surface. Since the understory consisted of vegetation with leaves instead of needles, the parametrisation method for the understory vegetation was considered the same as that for broad-leaved species. In order to secure a more accurate representation of canopy wetness which was also relevant to the calculation of the layer-specific conductance for $O_3$, RH values inside the canopy were calculated from the measured absolute humidity and simulated air temperature.

In addition, to secure a realistic simulation of $O_3$ in a column model like SOSAA we also forced the model's $O_3$ concentration at 23 m to resemble the observed value every time step, the $O_3$ concentration at other levels were then calculated by Eq. 17. In this way, we implicitly added the role of advection in determining the $O_3$ concentration above the canopy. The gap-filled observed values which were used for the forcing are shown in Fig. 2b.

     Several sensitivity cases have been conducted in this study (Table 1). In the case BASE all the parameters and methods were
kept the same as described in section 2.4. In cases RSOIL200, RSOIL600 and RSOIL800 $r_{soil}$ was altered to 200 s m$^{-1}$, 600 s m$^{-1}$ and 800 s m$^{-1}$, respectively. In the case FREEO3, the $O_3$ concentration at 23 m was computed from Eq. (17) instead of being set to the measurement data.

### 2.6   Sensitivity analysis of $r_{soil}$

$r_{soil}$ varied in different studies, ranging from 10 to 180 s m$^{-1}$ for dry soil and 180 to 1100 s m$^{-1}$ for wet soil (Massman, 2004).
In this study the dry deposition module was developed on the basis of the model from Ganzeveld and Lelieveld (1995) in which $r_{soil}$ was 400 s m$^{-1}$. In order to assess the uncertainties involved in estimating $r_{soil}$, different values of $r_{soil}$ ranging from 200

**Table 1.** Table of sensitivity cases. The case names and their short description texts are shown.

| name | description |
|------|-------------|
| BASE | the same as described in section 2.4 |
| RSOIL200 | $r_{soil} = 200$ s m$^{-1}$ |
| RSOIL600 | $r_{soil} = 600$ s m$^{-1}$ |
| RSOIL800 | $r_{soil} = 800$ s m$^{-1}$ |
| FREEO3 | $O_3$ concentration at 23 m was also computed instead of using observed data |

**Table 2.** The average (MEAN) and standard deviation (STD) of modelled and measured $O_3$ fluxes ($\mu$g m$^{-2}$ s$^{-1}$) above the canopy during different time periods (ALL for the whole month, D for daytime, N for nighttime) for different cases (OBS for measurement, BASE for basic settings used in this study, RSOIL200 uses the same settings as in BASE except $r_{soil} = 200$ s m$^{-1}$, similarly, RSOIL600 with $r_{soil} = 600$ s m$^{-1}$ and RSOIL800 with $r_{soil} = 800$ s m$^{-1}$) are shown. The relative error (RE) of modelled $O_3$ flux compared to the observation $(F_{t,mod} - F_{t,obs})/F_{t,obs}$ is also presented.

| CASES | ALL | | D | | N | |
|-------|-----|-----|-----|-----|-----|-----|
| | MEAN±STD | RE | MEAN±STD | RE | MEAN±STD | RE |
| OBS | $0.246 \pm 0.175$ | | $0.334 \pm 0.165$ | | $0.103 \pm 0.073$ | |
| RSOIL200 | $0.286 \pm 0.173$ | +16.4% | $0.375 \pm 0.162$ | +12.1% | $0.140 \pm 0.067$ | +35.0% |
| BASE | $0.250 \pm 0.153$ | +1.77% | $0.329 \pm 0.143$ | -1.74% | $0.120 \pm 0.059$ | +16.2% |
| RSOIL600 | $0.231 \pm 0.144$ | -6.00% | $0.305 \pm 0.134$ | -8.85% | $0.109 \pm 0.057$ | +5.16% |
| RSOIL800 | $0.219 \pm 0.139$ | -10.8% | $0.290 \pm 0.129$ | -13.2% | $0.101 \pm 0.055$ | -2.17% |

to 800 s m$^{-1}$ were tested in this study (Table 2). As can be expected, the modelled $O_3$ fluxes decrease as $r_{soil}$ increases. The BASE case shows the best performance in general, although it overestimates $\sim 16\%$ nighttime $O_3$ fluxes. Since the RSOIL200 case overestimates $O_3$ fluxes by $\sim 17\%$ in average for the whole month, $\sim 12\%$ at daytime and $\sim 35\%$ at nighttime, the RSOIL200 sensitivity case indicates that using this lower estimate, a value that might be more appropriate for high organic (and dry) soils, seems not to represent properly the role of soil removal at this site. On the other hand, taking higher resistance values, e.g., one of 600 or 800 s m$^{-1}$ seems to result in a better simulation of the role of the soil uptake at nighttime. However, considering the overall performance and better estimation of daytime $O_3$ fluxes, we used 400 s m$^{-1}$ as the soil resistance in this study.

## 3 Results and discussion

### 3.1 Micrometeorology

The simulated month was warm and dry with little precipitation. Moreover, the temperature decreased dramatically in the middle of the month. In the first half of month (Aug 1st to Aug 15th) the average temperature at 23 m was 19.0 °C, while it dropped to 12.1 °C in the second half of month (Aug 16th to Aug 31st) (Fig. 2a). Analysis of the full temperature record indicated that this transition in the weather conditions at the site was well simulated by the model. RH varied inversely with air temperature. Its average value increased only slightly from 66.0% in the first half of the month to 69.3% in the second half. However, a dramatic increase of daily mean RH values from 49.3% to 73.5% occurred between Aug 20th and Aug 21st (Fig. 2a). The combination of the dry weather and the large variation of temperature provided a good sample for verifying the $O_3$ dry deposition module.

Figure 3 shows the comparison results between simulated and measured horizontal wind speed and friction velocity ($u_*$). Both of them are essential for estimating the turbulent transport above and within the canopy as well as for the calculation of the quasi-laminar boundary layer resistance of leaves ($r_b$) at each canopy layer and the soil boundary layer resistance ($r_{bs}$). Figure 3a shows the good agreement between modelled and measured monthly-mean horizontal wind speed profiles during both daytime and nighttime. The wind speed decreases quickly inside the canopy due to canopy drag, then changes little below $0.5\,h_c$ until near the surface where wind speed varies logarithmically to zero on the surface. The model reproduces the diurnal cycle of $u_*$ but overestimates the nighttime values by $\sim 0.05$ m s$^{-1}$ in average above the canopy (Fig. 3c). Below the canopy crown at $\sim 3$ m, $u_*$ is underestimated by $\sim 0.02$ m s$^{-1}$ at nighttime and $\sim 0.05$ m s$^{-1}$ at daytime (Fig. 3d). The discrepancy is likely due to the limitation of representing the real heterogeneous dynamics by a 1D model with homogeneous canopy configuration.

### 3.2 PAR above and below the canopy crown

Photosynthetically active radiation (PAR) plays an important role in stomatal exchange which determines to a large extent the daytime vegetation uptake. The PAR above the canopy is calculated directly from the measured incoming short wave radiation serving as input to the model, and shows a daytime maximum of about $250 - 300$ W m$^{-2}$ during the simulation month. The PAR inside the canopy is calculated by considering the absorption, reflection and scattering effects of canopy leaves in the model (Sogachev et al., 2002). The comparison between modelled and observed PAR at $\sim 0.6$ m below the canopy crown is shown in Fig. 4. The monthly-mean diurnal cycle of attenuated PAR below the canopy crown in the model is consistent with the observation except two missing peaks at daytime (Fig. 4b). These two peaks in the measurement are the consequence of direct exposure of PAR sensors to incoming solar radiation. Such situation always occurs when point-wise measurement is compared with a model assuming a homogeneous forest canopy.

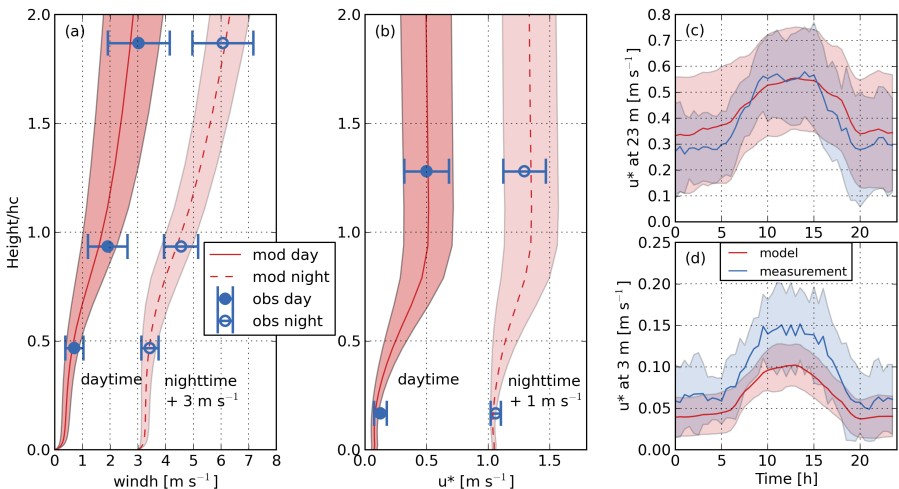

**Figure 3.** Modelled (red solid line for daytime, red dashed line for nighttime) and measured (blue solid circle for daytime, blue empty circle for nighttime) profiles of horizontal wind speed (windh) (a) and friction velocity ($u*$) (b). Nighttime values are shifted by 3 and 1 m s$^{-1}$ for wind and $u*$ for clarity of presentation, respectively. The ranges of $\pm 1$ standard deviation of modelled and measured data are marked as shades and error bars. The height is normalised by canopy height $h_c$. The monthly-mean diurnal cycles of modelled (red) and measured (blue) friction velocity at 23 m and 3 m are shown in (c) and (d). The ranges of $\pm 1$ standard deviation are marked as shades in the same colours.

## 3.3 Energy balance

The monthly-mean diurnal cycles of sensible heat flux, latent heat flux, net radiation and soil heat flux are shown in Fig. 5 in order to verify the simulated energy balance above the canopy. The upward energy flux or the loss of surface energy is represented by positive values. During daytime, the soil and canopy loses energy by heat fluxes and gains energy mainly from

5   net incoming solar radiation. At night, the surface loses energy by net upward long wave radiation with an average rate of $\sim$ 33 W m$^{-2}$, which is partly compensated by $\sim$ 20 W m$^{-2}$ downward sensible heat flux.

    During the simulation period the modelled diurnal cycles of energy fluxes agree well with the observation, although, for example, the latent heat flux is slightly underestimated by $\sim$ 30 W m$^{-2}$ during daytime. In the afternoon from 14:00 to 20:00 the sensible heat flux is underestimated by $\sim$ 20 W m$^{-2}$. This could be explained by the underestimation in net radiation. However,

10   the modelled values are generally within the one standard deviation range of the observations. The agreement between modelled and measured latent heat flux also indicates that the stomatal exchange, which controls the latent heat flux and is directly related to the stomatal resistance of $O_3$ and many other gaseous compounds, is realistically simulated as a function of the meteorological drivers.

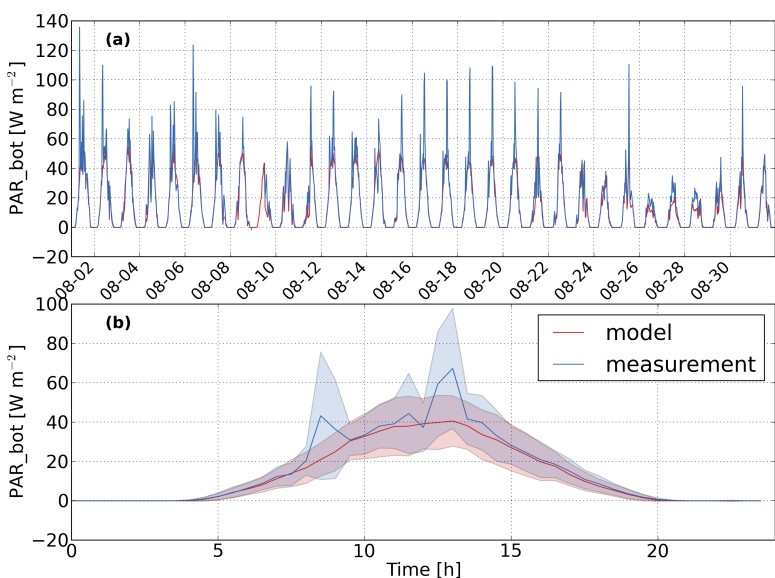

**Figure 4.** (a) Time series of PAR at 0.6 m from model (red) and measurement (blue) in August, 2010. (b) The monthly averaged diurnal cycle of time series in (a) for model (red) and measurement (blue). The range of $\pm 1$ standard deviation is marked by the shade with the same colour.

## 3.4   O$_3$ fluxes

The modelled time series of O$_3$ turbulent flux and its diurnal cycle are compared with the measurement data above the canopy (Fig. 6). In general, the modelled flux shows a good agreement with the observations especially in the second half of month (Fig. 6a). Large discrepancies mostly occur in the first half of month which is warm and dry. On the first 3 days of the month,

the O$_3$ turbulent flux is overestimated by the model. On some days at noon (e.g., Aug 9th, 12th, 13th, 14th, 27th, 30th), the model is not able to predict the observed high peaks of O$_3$ turbulent fluxes. In an average diurnal cycle of O$_3$ turbulent flux, the model does not capture the rapid increase of downward O$_3$ turbulent flux in the morning, but it follows the measurement well after 10:00. In general the agreement between the simulated and measured monthly-mean diurnal cycles of O$_3$ turbulent fluxes is promising.

Figure 7 shows the correlation between the simulated and measured O$_3$ turbulent fluxes above the canopy for different humidity conditions at daytime and nighttime separately. The overall $R^2$ between the modelled and measured O$_3$ turbulent fluxes for the whole dataset is 0.47. Among the four individual datasets under different conditions, the best prediction by the model occurs for the NH data points with $R^2$ of 0.37, followed by the results reflecting the daytime high humidity conditions ($R^2$=0.19). Note that these conditions with highest correlations are also the conditions with high relative humidity, especially

at nighttime. All the correlations are significant ($p < 0.001$) except the condition NL for which $R^2$ is only 0.02 (Fig. 7).

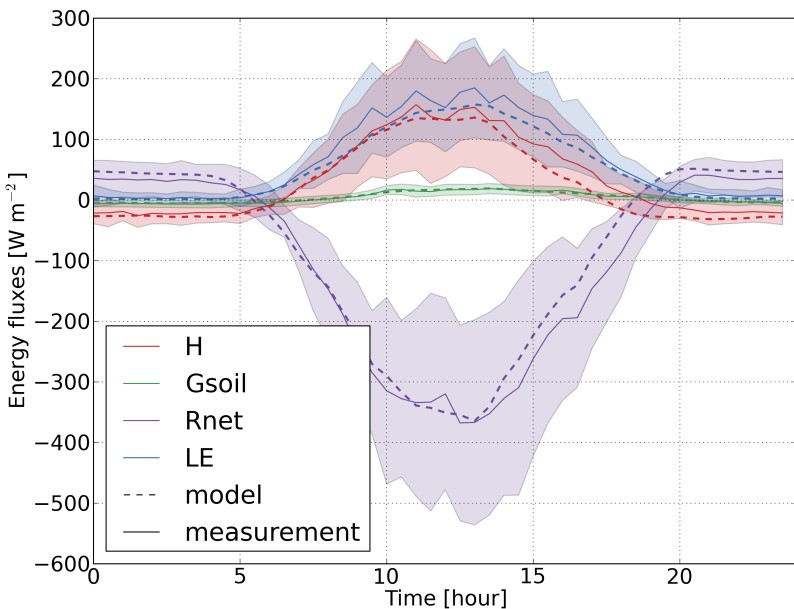

**Figure 5.** The monthly averaged diurnal cycle of different energy flux terms at 23 m above the ground for model (dashed lines) and measurement (solid lines), including sensible heat flux (H, red line), soil heat flux (Gsoil, green line), upward net radiation (Rnet, purple line, note the observed Rnet is at 67 m), latent heat flux (LE, blue line). The range of $\pm 1$ standard deviation for measurement data is plotted for every term by the shade with the same colour.

This indicates the difficulty of simulating the $O_3$ turbulent flux in weak turbulence at nighttime. Usually at nighttime RH is larger than 70% (Fig. 2), under this condition (NH condition), the wet skin uptake contributes more than 50% (Table 3) to the deposition flux. Therefore, the turbulent mixing above the ground which affects the deposition flux onto soil only plays a minor role on the deposition flux above the canopy. However, in NL condition which does not happen frequently, nearly
5   all the deposition inside the canopy is caused by soil deposition. Hence, the difficulty of simulating the exchange processes near the surface may cause more uncertainty of simulating the deposition flux onto soil surface under NL condition than NH condition. Moreover, the vertical advection of $O_3$ could also affect the turbulent flux at nighttime (Rannik et al., 2009), which complicates the analysis. On the other hand, there are only 69 observed data points in the condition NL which implies larger random uncertainty. When the surface is wetter, the simulated nocturnal $O_3$ turbulent fluxes correlate much better with the
10  measurement. In addition, the measurement data show a larger range of variation (about -1.2 – 0.0 $\mu$g m$^{-2}$ s$^{-1}$) compared to the range in the modelled $O_3$ turbulent flux (about -0.8 – 0.0 $\mu$g m$^{-2}$ s$^{-1}$), which implies that the model does not capture the $O_3$ turbulent flux peaks or the measurements are more scattered due to random errors. Regarding the low $R^2$ values here, we should consider the uncertainty of measured fluxes. Such uncertainty contributes to the data scattering when comparing the modelled and measured fluxes, such as in Fig. 7, and reduces the correlation statistics.

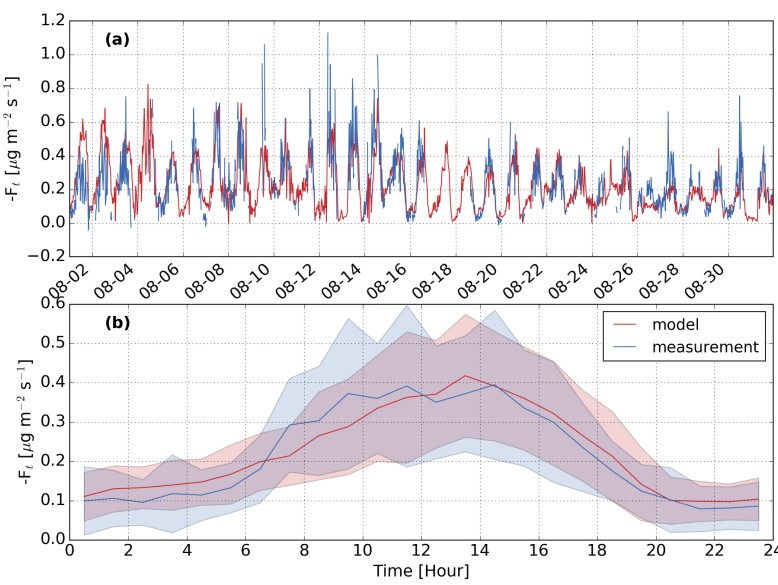

**Figure 6.** (a) Time series of the simulated (red) and measured (blue) $O_3$ turbulent fluxes above the canopy in August, 2010. (b) The monthly averaged diurnal cycles of time series presented in (a) for the model (red) and measurement (blue). The ranges of $\pm 1$ standard deviation are marked by the shades with the same colours. Positive values represent downward fluxes.

In general, the parametrisation of wet skin fraction (Eq. (15)) and its impact on $O_3$ non-stomatal removal seems to represent the $O_3$ deposition mechanisms inside the canopy well considering the good performance under high humidity conditions. Although the prediction of $O_3$ turbulent flux with weak turbulence at night under low humidity condition still has large uncertainties (Fig. 7), the simulated average nocturnal $O_3$ turbulent flux above the canopy shows a good agreement with the observation (Fig. 6b).

## 3.5 $O_3$ concentration profile

In order to assess if the good agreement between the observed and simulated $O_3$ turbulent fluxes above the canopy also implies a realistic representation of the $O_3$ concentration inside the canopy, we have conducted an evaluation of the simulated in-canopy $O_3$ concentration profile. The one-month averaged $O_3$ concentration profiles from model results and measurements are shown in Fig. 8. The large variation range results from the meteorological variations in this month, especially the abrupt transition in the middle of the month (Fig. 2). The average $O_3$ concentration of the whole month is 60.4 $\mu$g m$^{-3}$ at 23 m, then decreases gradually inside the canopy to 54.1 $\mu$g m$^{-3}$ at 4.2 m due to the in-canopy sinks. Similar vertical gradients are also found for the four different conditions. At night, the turbulent mixing is weaker compared to daytime which inhibits the downward transport

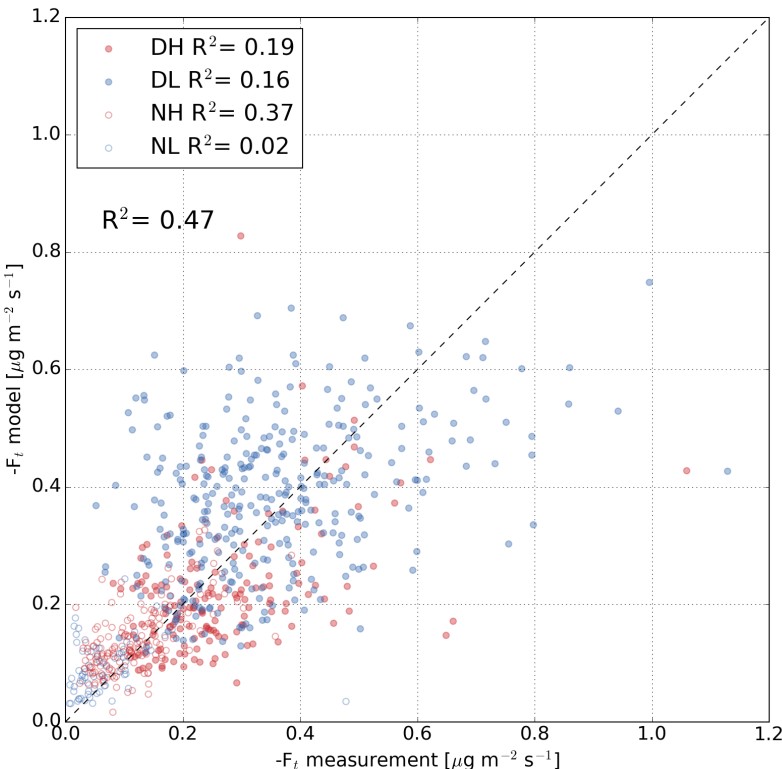

**Figure 7.** Scatter plots of modelled versus measured $O_3$ turbulent fluxes above the canopy. The data points are plotted separately for different groups (DH, DL, NH and NL) with their $R^2$ values shown in the legend. $R^2$ of the whole dataset is shown below the legend.

of air with larger concentration of $O_3$ into the canopy. Hence the $O_3$ removal by canopy and especially by soil surface results in larger gradient of $O_3$ inside the canopy during nighttime (Fig. 8).

The model results of $O_3$ concentration profiles show a good agreement with the observations except the slight overestimation for the DH condition below $\sim 8$ m ($0.45\ h_c$) and the apparent underestimation for the NL condition throughout the whole

5    canopy. This is consistent with the model results of the $O_3$ turbulent fluxes, which show $\sim 20\%$ underestimation for the DH condition and $\sim 38\%$ overestimation for the NL condition. In addition, the modelled vertical gradient of $O_3$ concentration during nighttime at drier conditions (NL) is much larger inside the canopy compared to the measured gradient, which implies that the soil deposition is largely overestimated when the soil and dry vegetation surface uptake dominates the overall removal inside the canopy. This also indicates that further investigation is needed for the more accurate representation of ground surface

10   deposition at different humidity conditions, including possibly the roles of uptake by the moss layer and soil humus layer.

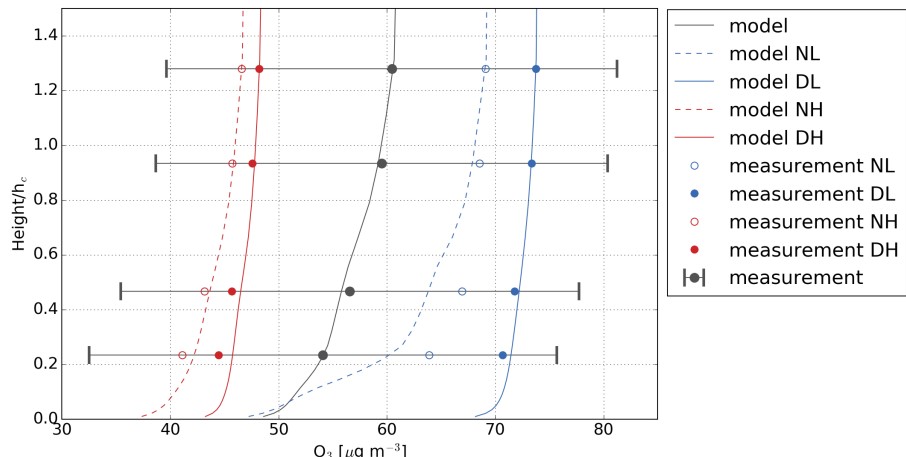

**Figure 8.** Measured average vertical profiles of $O_3$ concentration for the whole month (dark grey, the horizontal bars are $\pm 1$ standard deviations) and individual conditions (daytime under high humidity condition, labelled as DH with red filled circle; daytime under low humidity condition, labelled as DL with blue filled circle; nighttime under high humidity condition, labelled as NH with red empty circle; nighttime under low humidity condition, labelled as NL with blue empty circle). Modelled results are plotted as solid lines (daytime) and dashed lines (nighttime) with the same colour as measurement. The height is normalised by the canopy height $h_c$.

## 3.6 $O_3$ flux profile

The normalised cumulative $O_3$ deposition flux at layer $i$ can be obtained as

$$F_{c,i} = \frac{\sum_{k=1}^{i} F_k}{\sum_{k=1}^{N} F_k} \tag{19}$$

where $F_k$ is the $O_3$ deposition flux at layer $k$ and N is the layer index just above the canopy. The profiles of $F_c$ and the
contributions of different deposition pathways for four different conditions are shown in Fig. 9. For the whole month, the $O_3$ uptake is dominated by soil deposition below $0.2\,h_c$ ($\sim 3.6$ m) with only $\sim 8\%$ contribution from the understory vegetation via stomatal uptake. From $0.2\,h_c$ to $0.8\,h_c$ ($\sim 14.4$ m) the cumulative uptake on leaf surfaces increases with height due to dense leaves in the plant crown area. Above $0.8\,h_c$ the remaining small portion of biomass ($\sim 7\%$) provides less than 2% $O_3$ uptake compared to the total $O_3$ deposition.

The soil uptake contributes to the total $O_3$ deposition flux at both daytime and nighttime (Figs. 9b and 9c) with a percentage of $\sim 32\%$ and $\sim 54\%$, respectively. At daytime, $\sim 63\%$ of the $O_3$ deposition flux is due to stomatal uptake. While at nighttime, when RH is larger than 70% at most of the time, the cumulative wet skin uptake contributes $\sim 41\%$ to the total $O_3$ deposition. At nighttime under high humidity conditions, the wet skin uptake even contributes $\sim 51\%$ to the total $O_3$ deposition fluxes (Table 3). This indicates that wet skin uptake plays a crucial role at night which is consistent with the results in Rannik et al.
(2012). As a result, the simulated averaged non-stomatal contribution to the integrated $O_3$ deposition flux above the canopy is $\sim 37\%$ during daytime and $\sim 96\%$ during nighttime (Table 3). It should be noted that the stomata are not completely closed

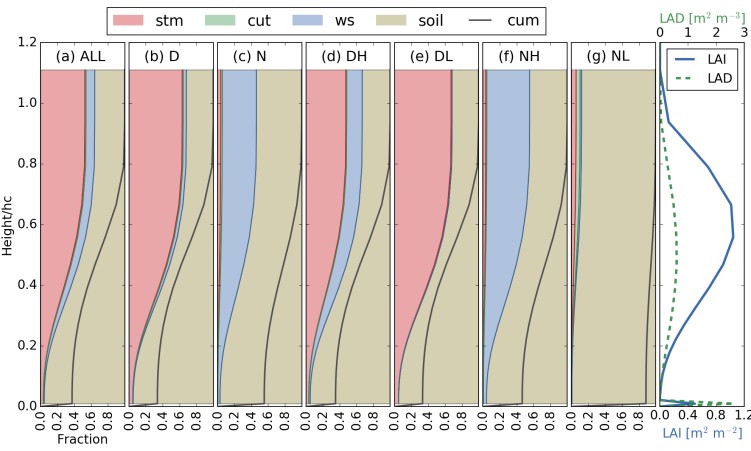

**Figure 9.** Simulated vertical profiles of cumulative $O_3$ deposition flux normalised by the integrated $O_3$ deposition flux above the canopy (cum, solid black line) for four conditions DH (a), DL (b), NH (c) and NL (d). D and N represent daytime and nighttime, H and L represent high and low humidity, respectively. Shaded areas are the cumulative contribution fractions for different deposition pathways, including stomatal uptake (stm, red), cuticle uptake (cut, green), wet skin uptake (ws, blue) and soil uptake (soil, pale brown). The all-sided LAI profile for each layer and LAD is plotted again here (e). The height is normalised by the canopy height $h_c$.

at night (Caird et al., 2007) and the minimum stomatal conductance at nighttime is about 5% of its maximum value at daytime (Kolari et al., 2007) which is similar with the simulation result here ($3.7\%/63.0\% \approx 6\%$, Table 3).

    Above 0.2 $h_c$, the stomatal uptake (DL, Fig. 9b), wet skin uptake (NH, Fig. 9c) or both of them (DH, Fig. 9a; NL, Fig. 9d) start to play a significant role in the cumulative $O_3$ deposition fluxes. Hence at 0.8 $h_c$ the cumulative contribution of soil
deposition is less than 50% except in the NL condition when both the cumulative stomatal uptake and wet skin uptake are limited. In all the four conditions the dry cuticle uptake is minor with a maximum contribution of about 5.0% for the NL condition.

    During daytime the sub-canopy layer, including soil surface, contributes about 38% to the integrated $O_3$ deposition, which is consistent with the results from Launiainen et al. (2013) in which the sub-canopy (lower than 4.2 m) contribution was 35–
45% at daytime. At night the contribution increases to around 60% due to the closed stomata in crown layers. This is much higher than that ($25 - 30\%$) in Launiainen et al. (2013) (Table 3). The overestimation could result from the underestimation of the soil resistance, which is difficult to determine in such a complex ground ecosystem. However, among these four different conditions with the same constant soil uptake efficiency, only under the nocturnal dry conditions (NL) there is apparently an overestimation in $O_3$ uptake and consequently underestimation of the $O_3$ concentration inside the canopy (Fig. 8). Therefore,
we expect that the poor performance for the NL condition also results from the limited data amount under this condition (only 69 data points) which leads to larger ratio of random uncertainty and thus smaller $R^2$.

    Moreover, the assumption that the resistance $r_{ac}$ between the understory vegetation and ground is not a limiting factor for soil deposition might not hold under certain conditions. On the other hand, Launiainen et al. (2013) studied one month earlier

**Table 3.** The first four columns are the contribution fractions of different deposition pathways (stm as stomatal uptake, wet as wet skin uptake, cut as cuticle uptake, soil as soil surface uptake) in the integrated $O_3$ deposition flux inside the canopy in the model. The last column is the sub-canopy (below 4.2 m) $O_3$ turbulent flux ($F_{t,mod}$(4.2m)) compared to the $O_3$ turbulent flux above the canopy ($F_{t,mod}$) in the model. Different conditions are listed along the row. D and N represent daytime and nighttime, H and L represent high and low humidity, respectively. ALL is for the whole dataset.

|  | stm | wet | cut | soil | $F_{t,mod}$(4.2m)/$F_{t,mod}$ |
|---|---|---|---|---|---|
| D | 63.0% | 3.79% | 1.12% | 32.1% | 38.0% |
| N | 3.70% | 40.5% | 1.87% | 53.9% | 59.5% |
| DH | 47.2% | 18.5% | 0.94% | 33.4% | 39.6% |
| DL | 67.1% | 0.00% | 1.17% | 31.8% | 37.6% |
| NH | 3.28% | 51.0% | 1.04% | 44.7% | 51.4% |
| NL | 5.42% | 1.78% | 4.73% | 88.1% | 89.5% |
| ALL | 52.5% | 10.4% | 1.25% | 35.8% | 41.7% |

period (July 1st to August 4th, 2010) than the time period (August 1st to August 31st, 2010) in this study, so the difference between these two studies could also be due to the meteorological and biological variations during the two summer months. However, the daytime contribution of the sub-canopy layer is consistent, so the difference between the two months could only play a minor effect.

## 3.7 Contribution of air chemistry

The role of chemical processes in explaining the $O_3$ removal inside the forest canopy has been discussed in previous studies (e.g., Altimir et al., 2006; Wolfe et al., 2011; Rannik et al., 2012; Launiainen et al., 2013). A study by Wolfe et al. (2011) found that the non-stomatal uptake over a Ponderosa pine stand in the US was associated with additional very reactive BVOCs being present besides the identified ones. On the other hand, Rannik et al. (2012) suggested that the air chemistry provided only minor contribution at SMEAR II. In order to estimate the contribution of chemical removal at SMEAR II, two different studies applied multi-layer models (Rannik et al., 2012; Launiainen et al., 2013) to simulate the $O_3$ fluxes and concentration inside the boreal forest canopy. However, both of them showed their limitations on estimating the chemical contribution. Rannik et al. (2012) only considered one chemical reaction of $O_3$ with $\beta$-caryophyllene. While in Launiainen et al. (2013), they simplified the chemical production and loss of $O_3$ with only two parameters to represent the first-order kinetic sink and photo-chemical production. In this study, we implemented a chemistry module with a detailed list of chemical reactions (see section 2.4.1), which was able to provide a more accurate estimation of chemical removal of $O_3$ inside the canopy.

In order to get rid of the effect of synoptic-scale transport of $O_3$ and only focus on the local sinks and sources, we applied the simulation case FREEO3. In this simulation case we ignored the role of advection and only considered the role of local sources and sinks inside the canopy, i.e., dry deposition, chemical production and loss, and turbulent transport. Here the time

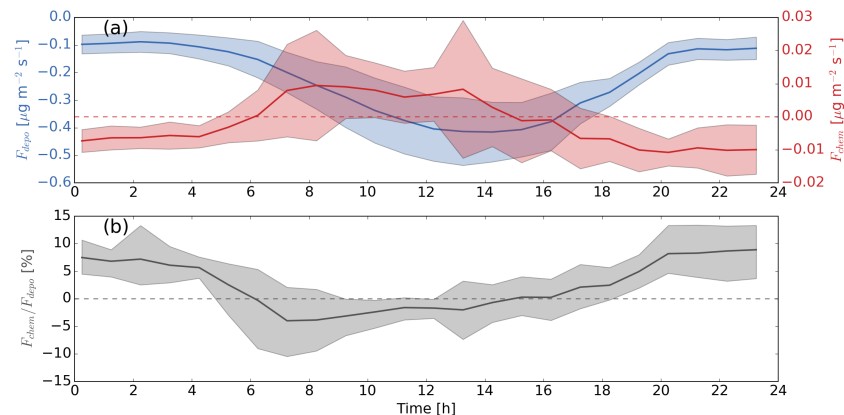

**Figure 10.** (a) The daily averaged (from Aug. 5th to 14th) production and loss caused by chemistry ($F_{chem}$, red) and dry deposition ($F_{depo}$, blue). (b) The ratio between $F_{chem}$ and $F_{depo}$. Zero lines for $F_{chem}$ and the ratio are plotted as dashed lines. Shaded areas show the range of $\pm 1$ standard deviation.

period from Aug. 5th to 14th were selected from the simulation results to analyse the local chemical contribution, because the modelled $O_3$ concentration fitted to the measurement the best during this period out of the whole month for the case FREEO3, which indicated that the advection also did not have apparent effect on the local observed $O_3$ variation. The daily averaged (from Aug. 5th to 14th) production and loss of total $O_3$ inside the canopy per square meter caused by dry deposition ($F_{depo}$) and

chemistry ($F_{chem}$) are plotted in Fig. 10. Positive values correspond to $O_3$ production and negative values represent $O_3$ loss. Here the chemistry production is a net effect of $O_3$ loss reactions and photo-chemical production. $F_{depo}$ (obviously negative) shows a maximum $O_3$ loss rate at about 14:00. The chemistry produces $O_3$ from morning at $\sim$ 06:00 to the afternoon at $\sim$ 15:00, and destroys it throughout the other time of the day, especially at nighttime (Fig. 10). The ratio between $F_{chem}$ and $F_{depo}$ shows that chemical removal has its largest contribution of $\sim$ 9% of the dry deposition sink in average at nighttime from

20:00 to 04:00. At daytime, our model simulation indicates that the $O_3$ production caused by chemistry can compensate up to $\sim$ 4% of dry deposition loss in average. However, during the selected period, the chemical contribution and compensation can reach up to $\sim$ 24% and $\sim$ 20% at most. This indicates that in general chemistry has minor impact on $O_3$ alteration, but at some specific time the chemical production and removal of $O_3$ can still play a significant role.

As a comparison, we also calculated the time scales of different removal processes to estimate the contribution of air chem-

istry. The average value of measured $O_3$ flux ($F_{O_3,avg}$) in August, 2010 above the canopy was 0.33 $\mu$g m$^{-2}$ s$^{-1}$ at daytime and 0.10 $\mu$g m$^{-2}$ s$^{-1}$ at nighttime whereas the $O_3$ concentration ([$O_3$]) inside the canopy was about 61.6 $\mu$g m$^{-3}$ during daytime and 50.5 $\mu$g m$^{-3}$ at night. So the time scale of total $O_3$ flux ($\tau_{O_3}$) could be obtained from

$$\tau_{O_3} = [O_3]\, h_c / F_{O_3,avg} \tag{20}$$

which was $\sim 3400$ s ($\sim 1$ h) for daytime and $\sim 9100$ s ($\sim 2.5$ h) for nighttime. On the other hand, the total $O_3$ reactivity ($y$) at 18 m during a similar time period and at the same boreal forest station was calculated by Mogensen et al. (2015), which was $1.58 \times 10^{-5}$ s$^{-1}$ and $1.67 \times 10^{-5}$ s$^{-1}$ for noon and 2 a.m. at night. If the same values were assumed to be applicable also inside the canopy, the time scale of the $O_3$ removal by chemistry ($\tau_{c,O_3}$),

$$\tau_{c,O_3} = y^{-1}, \tag{21}$$

was $\sim 63300$ s ($\sim 18$ h) for daytime and $\sim 59900$ s ($\sim 17$ h) for nighttime. These estimates showed that the chemical removal accounted for about 5% (3400/63300) and 15% (9100/59900) of the total $O_3$ removal within the canopy at daytime and nighttime, respectively.

Compared to the simulation results, the time scale analysis could not reflect the photochemical production of $O_3$ during daytime, hence the estimation of net chemical effects is not possible with this method. For nighttime, the time scale analysis overestimates the average contribution of chemical removal by about 88% (15% compared to 8%, 8% is obtained from 9%/(100%+9%)). The comparison result could act as a proof of the statement in Wolfe et al. (2011), which argued that the time scale might not be a good criteria of chemical influence.

## 4 Summary

A detailed multi-layer $O_3$ dry deposition model has been implemented into SOSAA to investigate the $O_3$ uptake by canopy and soil surface at a boreal forest station SMEAR II. The presented detailed analysis of the $O_3$ deposition processes for this site also quantified various removal processes, e.g., by the dry and wet cuticle, by stomatal uptake and by the soil surface.

In this model the fraction of wet skin on canopy leaves was parametrised according to RH values to analyse the potential role of canopy wetness on $O_3$ deposition for both high and low humidity conditions. Moreover, the multi-layer model also enabled the study of deposition processes inside the canopy and the partitioning of $O_3$ deposition fluxes between the canopy crown and sub-canopy. In this study, the model has been validated by comparing the modelled and measured $O_3$ turbulent flux above the canopy and its concentration profile inside the canopy.

Further investigation has been done through a more in-depth correlation analysis on $O_3$ turbulent fluxes for nighttime and daytime under high and low humidity conditions. The simulated $O_3$ turbulent fluxes above the canopy correlated reasonably well with the measurement for the whole month with $R^2$ of 0.47 ($p < 0.001$), which was also consistent with the plausible prediction of $O_3$ concentration profile inside the canopy. The significant correlation ($p < 0.001$) also applied to the daytime humid and dry as well as nighttime humid conditions (DH, DL and NH) with $R^2$ of 0.19, 0.16 and 0.37. However, the model was not able to predict high peaks with $O_3$ turbulent fluxes larger than 0.8 $\mu$g m$^{-2}$ s$^{-1}$. The model also did not capture well the measured $O_3$ removal for the nocturnal dry condition (NL), in which $R^2$ was only 0.02 and the $O_3$ concentration inside the canopy was largely underestimated (Figs. 7 and 8). The main reason could be the uncertainty of simulating the exchange processes near the ground in weak turbulent condition at nighttime when the soil deposition dominated the deposition flux inside the canopy.

Nearly all of the $O_3$ uptake occurred below 0.8 $h_c$ inside the canopy. During daytime, the contributions of stomatal uptake ($\sim 47\%$), wet skin uptake ($\sim 19\%$) and soil uptake ($\sim 33\%$) were significant for the total $O_3$ uptake under high humidity conditions. While under low humidity conditions the stomatal ($\sim 67\%$) and soil uptake (32%) contributed dominantly the overall canopy deposition. During nighttime, the stomatal uptake contribution ($\sim 3\%$) was not zero, but was much smaller compared to the wet skin uptake ($\sim 51\%$) under high humidity conditions. For the low humidity condition at night, nearly all the deposition ($\sim 88\%$) was due to soil uptake. Since RH was larger than 70% at most of the time during night, the uptake by wet canopy could be a dominant factor for the nocturnal $O_3$ removal. In addition, the simulated non-stomatal contributions to the integrated $O_3$ deposition fluxes were estimated as about 53%, 33%, 97% and 95% for conditions DH, DL, NH and NL, respectively (Table 3).

The modelled contribution of sub-canopy deposition during daytime ($\sim 38\%$) was consistent with that ($35 - 45\%$) in Launiainen et al. (2013), but it was much higher at nighttime ($\sim 60\%$) compared to that in the same study ($25 - 30\%$) (Table 3). This discrepancy at nighttime was most likely due to the overestimation of soil uptake.

The contribution of $O_3$ removal by chemical reactions with currently identified BVOCs has also been evaluated. In general the air chemistry played a minor role in $O_3$ uptake inside the canopy. In the simulated averaged diurnal cycle, the air chemistry produced $O_3$ during daytime from about 06:00 to 15:00, compensating up to 4% of dry deposition sinks. While at nighttime, the chemical loss enhanced $O_3$ removal by $\sim 9\%$ of that by dry deposition. A qualitative estimation of chemical contribution with time scale analysis was also conducted as a comparison. However, this method overestimated the air chemical removal by about 88% for nighttime and it was not able to reflect the $O_3$ production at daytime.

This study is the first step to establish a detailed gas dry deposition model in SOSAA. Further analysis of dry deposition will be done for other chemical compounds, especially for BVOCs. This will improve not only the ability to simulate air chemistry and aerosol processes but also our understanding of the mechanisms involved in the removal processes at boreal forest. In addition, it is also of scientific interest to investigate how future climate change might ultimately affect the removal processes of compounds like $O_3$ and BVOCs for boreal forests.

## Appendix A: Table of symbols

Table 4: Table of symbols

| symbol | value | unit | description |
|--------|-------|------|-------------|
| $h_c$ | 18 | m | canopy height |
| LAI | | $m^2\,m^{-2}$ | integral all-sided leaf area index, it can also represent the LAI at each layer in the context |
| $T$ | | K | air temperature |
| $q_v$ | | $kg\,m^{-3}$ | specific humidity |
| RH | | - | relative humidity |

| symbol | value | unit | description |
|---|---|---|---|
| $X$ | | - | scalar quantity |
| $u_*$ | | m s$^{-1}$ | friction velocity |
| $u_{*g}$ | | m s$^{-1}$ | friction velocity near the ground |
| H | | W m$^{-2}$ | sensible heat flux |
| LE | | W m$^{-2}$ | latent heat flux |
| $F_{t,X}$ | | - | turbulent flux of $X$ |
| $F_t$ | | $\mu$g m$^{-2}$ s$^{-1}$ | O$_3$ turbulent flux |
| $K_t$ | | m$^2$ s$^{-1}$ | turbulent eddy diffusivity |
| $K_h$ | | m$^2$ s$^{-1}$ | turbulent eddy diffusivity for heat fluxes |
| TKE | | m$^2$ s$^{-2}$ | turbulent kinetic energy |
| $\varepsilon$ | | m$^2$ s$^{-3}$ | dissipation rate of TKE |
| $\omega$ | | s$^{-1}$ | specific dissipation of TKE |
| $C_{p,air}$ | 1009.0 | J kg$^{-1}$ K$^{-1}$ | latent heat flux |
| $\rho_{air}$ | 1.205 | kg m$^{-3}$ | air density |
| $\gamma_d$ | 0.0098 | K m$^{-1}$ | lapse rate of dry air |
| $L_v$ | $2.256 \times 10^6$ | J kg$^{-1}$ | latent heat of vapourisation for water |
| $C_\mu$ | 0.0436 | - | closure constant in calculating $K_t$ |
| $A_s$ | | m$^2$ m$^{-3}$ | soil area index |
| $Q_{chem}$ | | $\mu$g m$^{-3}$ s$^{-1}$ | chemical production and loss |
| $F$ | | $\mu$g m$^{-2}$ s$^{-1}$ | O$_3$ deposition flux |
| [O$_3$] | | $\mu$g m$^{-3}$ | O$_3$ concentration |
| $V_d$ | | m s$^{-1}$ | layer-specific conductance for O$_3$ |
| $V_{dveg}$ | | m s$^{-1}$ | layer-specific leaf surface conductance |
| $V_{dsoil}$ | | m s$^{-1}$ | soil conductance |
| $r_{veg}$ | | s m$^{-1}$ | leaf surface resistance |
| $r_{veg1}$ | | s m$^{-1}$ | leaf surface resistance to the side without stomata |
| $r_{veg2}$ | | s m$^{-1}$ | leaf surface resistance to the side with stomata |
| $r_b$ | | s m$^{-1}$ | quasi-laminar boundary layer resistance over leaf surface |
| $r_{ac}$ | 0 | s m$^{-1}$ | resistance of turbulent transport from the reference height of the understory vegetation to the soil surface |
| $r_{bs}$ | | s m$^{-1}$ | soil boundary layer resistance |
| $r_{soil}$ | 400 | s m$^{-1}$ | soil resistance |
| $r_{stm}$ | | s m$^{-1}$ | stomatal resistance |
| $r_{stm,\text{H}_2\text{O}}$ | | s m$^{-1}$ | stomatal resistance for water vapour |

| symbol | value | unit | description |
|---|---|---|---|
| $r_{mes}$ | 0 | s m$^{-1}$ | mesophyllic resistance |
| $r_{cut}$ | $10^5$ | s m$^{-1}$ | cuticle resistance |
| $r_{ws}$ | 2000 | s m$^{-1}$ | wet skin resistance |
| $f_{wet}$ | | - | fraction of wet skin |
| $D_{\mathrm{H_2O}}$ | $2.12 \times 10^{-5}$ | m$^2$ s$^{-1}$ | molecular diffusivity of water vapour |
| $D_{\mathrm{O_3}}$ | $1.33 \times 10^{-5}$ | m$^2$ s$^{-1}$ | molecular diffusivity of O$_3$ |
| $\kappa$ | 0.41 | - | von Kármán constant |
| $\delta_0$ | | m | the height above ground where the molecular diffusivity is equal to turbulent eddy diffusivity |
| $z_*$ | 0.1 | m | the height under which the logarithmic wind profile is assumed |
| Sc | 1.07 | - | Schmidt number for O$_3$ |

*Author contributions.* Putian Zhou implemented the deposition code into SOSAA, made the simulation runs, analysed the results and wrote the main part of this manuscript. Laurens Ganzeveld provided and developed the deposition code, suggested the concepts of manuscript structure, contributed to the micrometeorology part and the discussions related to O$_3$ fluxes. Üllar Rannik contributed to the micrometeorology part, the discussions related to O$_3$ flux measurements and the discussions in chemical removal processes. Luxi Zhou contributed to implementing the deposition code into SOSAA and configuration of simulation runs. Rosa Gierens contributed to the configuration of meteorology part in SOSAA and configuration of simulation runs. Ditte Taipale contributed to the discussions related to air chemistry and site description. Ivan Mammarella contributed to discussions related to O$_3$ flux measurements. Michael Boy provided SOSAA code and the main concept and structure of this manuscript.

*Acknowledgements.* This work was supported by Maj ja Tor Nessling funding, the Academy of Finland (projects 1118615 and 272041), CRAICC (Cryosphere-atmosphere interactions in a changing Arctic climate), eSTICC (eScience tools for investigating Climate Change in Northern High Latitudes) and FCoE (The Centre of Excellence in Atmospheric Science - From Molecular and Biological processes to The Global Climate). This work was also supported by institutional research funding (IUT20-11) of the Estonian Ministry of Education and Research, and the European Regional Development Fund (Centre of Excellence EcolChange). The authors also wish to acknowledge CSC - IT Center for Science, Finland, for computational resources.

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
