# Peer review of "Simulating ozone dry deposition at a boreal forest with a multi-layer canopy deposition model"

_Atmospheric Chemistry and Physics, 2016_

## Referee Comment (RC1) · Anonymous Referee #1 · 25 May 2016

This paper presents an analysis of ozone deposition to a Scots pine forest based on a 1D multi-layer chemical transport model. The study includes an implementation of a resistance scheme for ozone deposition into an existing model, simulations for a one-month period and comparisons with measurement data. The results highlight the importance of non-stomatal sinks, both within the forest canopy and at the soil interface, and the role of surface wetness in mediating these fluxes. The topic of the study is highly relevant, as it becoming increasingly evident that ozone deposition to vegetation is controlled by interplay of processes that are difficult to describe by traditional 'big-leaf' models. In addition, the high-quality long-term data available provide an improved opportunity for development of more advanced surface exchange models.

The paper is fundamentally sound, although there are some issues that need further consideration. The measurement data from the SMEAR II site employed in the study

are excellent, and the SOSAA model serves as an advanced platform for studying surface exchange processes. For the most part, the paper is clearly written and based on sufficient analysis. I would recommend publishing the paper after a major revision that pays adequate attention to the following comments.

Major comments

The model description is incomplete:

(1) The authors state that they have implemented a multi-layer dry deposition model into SOSAA, which is a 1D chemical transport model. SOSAA is described in Section 2.3.1, which lists different modules and references but does not explain the model types or physical principles. The key elements of SOSAA relevant to the present study, especially turbulent mixing and the derivation of eddy diffusivity, should be described in more detail.

(2) Due to the incomplete model description, it is not obvious for the reader that the vertical mixing of O3 is calculated similarly to that of any other compound in SOSAA (Eqs. 5 and 6), and that the "multi-layer O3 deposition model" actually consists of the few resistance terms shown in Fig. 1 (of which not all are effective). As SOSAA has previously been used for simulating the exchange of reactive compounds and latent heat within a forest canopy, obviously it must include some sort of description of the surface exchange processes corresponding to stomatal uptake at least. This relationship should be explained, especially for the stomatal resistance of both overstory and understory vegetation.

(3) The presentation of input data should be clearer and specify the data actually used for the deposition calculations, i.e. which data are taken from measurements and what is derived within SOSAA. This is important, as a large part of the paper is dedicated to testing the modelled meteorological variables. For example, a comparison with observations is presented for u*, but it is not explained how the modelled profile is obtained or how it is utilised in the model.

[Figure]

(4) The description of O3 parameterisation (Section 2.3.2) presents parameter values with no justification or citation, or refers to model results that are not shown:

(4.1) r_soil is modified from a default value based on model simulations, which are not discussed. These simulations should be shown and would serve as a useful sensitivity test.

(4.2) r_ac is set to a very small arbitrary value. What is the point of including a resistance of 1 s m-1 in series with a resistance of 600 s m-1?

(4.3) r_b depends on molecular diffusivity and wind speed (or friction velocity, p.9). Please present the formula or an exact reference.

(4.4) r_stm is calculated from evapotranspiration rate in SOSAA. How?

(4.5) r_mes, r_cut and r_ws have constant values. Where do these come from?

(4.6) f_wet is a function of RH, for which the authors cite a grey literature report that does not even include original data on canopy wetness. Isn't there anything more substantial available?

There are some issues with the resistance scheme:

(5) An ineffective aerodynamic resistance term (r_ac) is included in series with the soil resistance (cf. Comment 4.2 above). However, a much more important term, namely the near-soil boundary layer resistance, is ignored in the model.

(6) The leaf surface resistance has a general formulation (Eq. 3) so as to represent both needle-shaped and broad leaves. This is accomplished by a scaling factor (alpha = 0.5) introduced to account for one-sided stomatal exchange on leaves. As the non-stomatal exchange takes place on both sides of such a leaf, with separate boundary layers, this scaling does not result in a correct formula for deposition on two-sided leaves. The authors describe alpha as a correction factor, so it may represent an approximation. However, this approximation should be justified.

Analysis of results could be improved:

(7) The authors demonstrate that the model performs well in high humidity conditions but fails during the night-time when RH is low. As wet needle surfaces require additional parameterisations (RH dependent resistance, wet surface fraction), it is surprising that the authors do no first try to develop a parameterisation that performs well in dry conditions, i.e. in a much simpler case. Instead, they refer to simulation and measurement problems due to weak turbulence, but do not explain how these would depend on RH. If the measurement uncertainties increase with weakening turbulence and affect the model validation (and the u* screening does not help), then this could be easily tested. As the soil resistance is given as a plausible explanation for the mismatch, it would also seem logical to test if a better fit can be obtained by varying this resistance (see Comment 4.1. above).

(8) Only ozone fluxes are considered in the analysis. As the modelled flux depends on the modelled concentration, which is affected by various processes and has a systematic diurnal cycle, it is difficult to assess how well the deposition processes are modelled by comparing fluxes alone. Perhaps you could have a look at the flux/concentration ratio (commonly called deposition velocity)?

(9) As the ozone fluxes measured at SMEAR II have been analysed in a large number of previous studies, including two different multi-layer models (Rannik et al., 2012; Launiainen et al., 2013), I would expect to see a more systematic comparison of these results. From the process modelling point of view, it would be useful to discuss how and why the modelled results differ between the three multi-layer models.

(10) The discussion of chemical removal (Sect. 3.7) is based on the reactivity estimates obtained from the literature. According to the model description, the SOSAA model employed here includes a detailed chemistry module, which I assume was used in the present simulations. Why are these calculations not utilised for estimating the importance of in-canopy chemistry?

The language should be improved:

(11) Even though I indicated in my access review that a linguistic revision is necessary, there are still numerous errors, some of which impair presentation. A few examples are given in the detailed comments below.

Minor comments

P1/L23: "under current knowledge of air chemistry" is obvious so can be removed.

P2/L24: Any more recent studies?

P2/L31-: A reference is needed for "the boreal forest emits a large portion of BVOCs". The examples discussed are for California.

P3/L9-12: Unclear logic. It is not only removal processes that are relevant. The introduction of eddy-covariance measurements to the discussion seems awkward. Please reformulate.

P3/L14: Wesely (1989) describes a single model, which is based on the big-leaf approach. So this sentence ("Among these models...") makes little sense.

P3/L14-15, "usually": Very often big-leaf models are used as inferential models.

P3/L15: Zhang et al. (2002) deal with deposition parameterisations rather than large-scale modelling.

P3/L18: Altimir et al. (2006) do not employ a multi-layer model.

P3/L23: A paper from 2000 is hardly suitable for evaluating recent models.

P3/L24, "process-based": Unclear which processes are referred to here. The implementation consists of a largely empirical resistance parameterisation.

P3/L33: Unclear which challenges are referred to here.

P4/L23: What is meant by "the same below"?

[Figure]

P4/L27-28: Why was the ozone flux calculated with data from a different anemometer than for other fluxes?

P4/L31: Did you correct the O3 flux data for high-frequency losses? How large were the corrections?

P5/L28: What does "partly constrained" mean?

P6/L11: I would recommend against using the term "deposition velocity" for layer-specific conductances.

P6/L19-20: What does "the unit is the same..." mean?

P6/L27-28: Unclear language; please rephrase.

P7/ Eq.5: This is a strange combination of partial derivatives and finite differences. Please present the equation in a mathematically consistent form. You also need to assume constant air density here. It would be more appropriate to present the 'flux' as mass flux density (g m-2 s-1).

P7/Eq.5: How did you solve for [O3]. If it is a common procedure within SOSAA, perhaps you could explain it in Sect. 2.3.1.

P8/L1-2: How did you do the forcing? Fig. 2b does not explain this.

P9/Table 1: u* is not limited to the canopy top.

P9/L13, P10/L1, P11/L7, "was calculated": How? These should be moved to the methods description.

P10/Fig. 2b: Gap-filling of data is not described in the paper. Why was it performed? For which variables?

P10/L8 (also elsewhere): These data are measured well above the canopy, so why are they referred to as "canopy top".

P11/Fig. 3, P13/Fig. 5: Are these data screened for low turbulence?

P12/L11-12: How do the low humidity conditions affect turbulent mixing, making this difficult to simulate?

P13/Table 2, P15/Fig. 7: Why is the R2 of the full data set higher than the R2 of any of the four subsets?

P14/L6: Can you estimate how much the correlation was affected by random uncertainty?

P15/Fig. 7: The caption is difficult to read.

P16/L11: The notation related to the cumulative flux is not obvious.

P16/L14: No stomatal contribution is indicated for the understory vegetation in Figure 9.

P16/L14-P17/L4: Unclear presentation. Does "uptake on leaf surfaces" refer to the flux or the cumulative flux (accumulated from the bottom)?

P17/Fig. 9: What explains the stomatal uptake during the night-time?

P17/L11: Please quantify the "limited O3 uptake", as it is obvious that small surface area corresponds to small uptake.

P17/L12-13: These percentage contributions only refer to the mean values of the four data sets, so discussion of variation may be misleading here.

P17/P13: This may be explained by Launiainen et al. (2013), but the meaning of the "sub-canopy layer" is unclear. Does it include some other vegetation surfaces in addition to the understory vegetation and soil?

P17/L14-17: The contributions cited from Launiainen et al. (2013) do not add up to 100%; why?

P18/L1: How was the soil resistance determined in the first place?

P18/L3-5: I do not see how this conclusion about EC measurements results from the
data presented here.

P19/L8-9: You should explain how these percentages were obtained.

P19/L24: No data on BVOC removal are presented in this study.

P20/L14: Poor presentation; please rephrase.

P20/L19-20: I do not see how these different flux partitionings would indicate "the difficulty of simulating and measuring O3 deposition at night".

Technical comments

P1/L18,L19: Incorrect grammar.

P2/L1-3: Unnecessary material for the abstract.

P3/L27-28: "manuscript in preparation" is not a useful reference.

P5/L10: Incorrect grammar.

P8/L8: Incorrect grammar.

P8/L13: Repetition from the introduction.

P9/L1 (also elsewhere): replace "showed" by "shows".

P12/L9,L13: Incorrect grammar.

P17/L17: Typo.

P20/L9: Incorrect grammar.

---

## Referee Comment (RC2) · Anonymous Referee #3 · 28 Jul 2016

The paper represents an effort to model ozone sinks into a boreal forest canopy. I appreciate the attempt to go beyond the traditional big leaf models in order to explain different O3 sinks driven by turbulence and energy balance at different levels. The article reads well and confirms previous finding that relevant stomatal sinks occurr during the day while chemical reactions with VOC are important mainly during the night time. However, I would have appreciated a more extended parameterization and a better description of the model in order to clearly understand the formalism adopted to predict energy balance terms. There are some arbitrary choices of parameters, and not a convincing analysis of sensitivity or results from a model calibration. A table showing results from a sensitivity analysis should be provided. Basic questions like: what could be the effect of an increase in air temperature and precipitation regimes on ozone deposition? Are not resolved, although it would have been nice triggering the

model for some predictions of Ozone deposition under future environmental changes. In general the paper lacks of more mechanistic explanations of the results, with more discussion on the possible drivers of dry and wet ozone deposition.

Pag 2 line 25: You mention again that dew on leaves can increase deposition, but could you spend two lines mensioning the reasons or hypothesis why a hydrofobic molecul reacts so fast on wet surfaces? Pag 3 line 10. What about NOx emitted from soils? Couldn't fast reactions between O3 and NO lead to high O3 fluxes in the sub-canopy region? Pag 3 line 34: Only one month to test the model? The relative contributions of O3 sinks changes a lot during the seasons in repsonse to air temperature and plant phenology. It is a pity that such an important modelling effort is limited to one month, I would extend to the all vegetative season. Pag 5 line 5: Extensive research has been conducted in Yuttiala to refine turbulence limitation to flux measurements. Why should we expect an ustar threshold different from other scalars measured at the site? Pag 6 line 20: do you have experience of subcanopy O3 fluxes so that you can better parameterize soil reisstances? It seems here that usage of one value rather than another is arbitrary and not properly calibrated. Pag 7 line 15. So you mean that Kt has been estimated form measured fluxes? Or in which other way? Reading through the manuscript I feel like the description of the model is not accurate, and more informations should be provided. Pag 19 line 15: Can you say that NOx are also not relevant in the boreal forest? Pag 20 line 11: Since the Stomatal resistance is calculated based on evapotranspiration, are you sure that relevant nocturnal soil evaporation does not contribute significantly to Rc? Have you tried to separate canopy transpiration form soil evaporation in the model?
* * *

---

## Author Comment (AC1) · 6 Oct 2016

**Reply to comments on "Simulating ozone dry deposition at a boreal forest with a multi-layer canopy deposition model"**

October 6, 2016

We would like to appreciate the reviewer for the detailed and valuable comments which helped us a lot to improve the manuscript. Our reply to all the comments are shown below.

1. **Comments: (1) Authors state that they have implemented a multi-layer dry deposition model into SOSAA, which is a 1D chemical transport model. SOSAA is described in Section 2.3.1, which lists different modules and references but does not explain the model types or physical principles. The key elements of SOSAA relevant to the present study, especially turbulent mixing and the derivation of eddy diffusivity, should be described in more detail.**

   Answer: We added a description of the turbulent mixing calculation in SOSAA and more details about the emission and chemistry as:

   "In SOSAA, the horizontal wind velocity ($u$ and $v$), temperature ($T$), specific humidity ($q_v$), turbulent kinetic energy (TKE) and the specific dissipation of TKE ($\omega$) are computed every time step (10 s) by prognostic equations. In order to represent the local to synoptic scale effects, $u$, $v$, $T$ and $q_v$ near and within the canopy are nudged to local measurement data at SMEAR II station with a nudging factor of 0.01. A TKE-$\omega$ parameterization scheme is used to calculate the turbulent diffusion coefficients ($K_t$) (Sogachev, 2009),

   $$K_t = C_\mu \frac{\text{TKE}}{\omega} \tag{1}$$

   $$\omega = \frac{\varepsilon}{\text{TKE}} \tag{2}$$

   where $\varepsilon$ is the dissipation rate of TKE and $C_\mu$ is a closure constant. Hence the turbelent flux of a quantity $X$ ($F_{t,X}$) can be computed as

   $$F_{t,X} = -K_t \frac{\partial X}{\partial z} \tag{3}$$

   where upward fluxes are positive and vice versa. Specifically, the sensible heat flux (H) and latent heat flux (LE) at each model layer are computed as

   $$\text{H} = -C_{p,air}\rho_{air}K_t \left( \frac{\partial T}{\partial z} + \gamma_d \right) \tag{4}$$

   $$\text{LE} = -L_v K_t \frac{\partial q_v}{\partial z} \tag{5}$$

   where $C_{p,air}$ (1009.0 J kg$^{-1}$ K$^{-1}$) is the specific heat capacity at constant pressure. $\rho_{air}$ (1.205 kg m$^{-3}$) is the air density which is a constant in the model. $\gamma_d$ (0.0098 K m$^{-1}$) is the lapse rate of dry air. $L_v$ (2.256 × 10$^6$ J kg$^{-1}$) is the latent heat of vaporization for water."

"The upper boundary values of $u$, $v$, $T$ and $q_v$ are constrained by the ERA-Interim reanalysis dataset provided by the European Centre for Medium-Range Weather Forecasts (ECMWF, Dee et al., 2011). At the canopy top, the incoming direct and diffuse global radiations measured at SMEAR II station, and the long wave radiation obtained from the ERA-Interim dataset are read in to improve the energy balance closure. Then the reflection, absorption, penetration and emission of three bands of radiation (long-wave, near-infrared and PAR) at each layer inside the canopy are explicitly computed according to the radiation scheme proposed by Sogachev et al. (2002). At the lower boundary, the measured soil heat flux at SMEAR II are used to further improve the representation of surface energy balance. All the input data are interpolated to match the model time for each time step. With the input data, the mass and energy exchange between atmosphere and plant cover (including the soil underneath) and the radiation attenuation inside the canopy are optimal to simulate the micrometeorological drivers of $O_3$ deposition at this site."

"In current SOSAA, a modified version of MEGAN has been used to simulate the emissions of BVOCs from the trees. The emissions of some important BVOCs are included, e.g., monoterpenes ($\alpha$-pinene, $\beta$-pinene, $\Delta^3$-carene, limonene, cineol and other minor monoterpenes (OMT)), sesquiterpenes (farnesene, $\beta$-caryophyllene and other minor sesquiterpenes (OSQ)) , 2-methyl-3-buten-2-ol (MBO). The chemistry mechanism is from MCMv3.2 including needed inorganic reactions and the full MCM oxidation paths for methane ($CH_4$), isoprene, MBO, $\alpha$-pinene, $\beta$-pinene, limonene and $\beta$-caryophyllene. We have also included the first-order oxidation reactions with OH, $O_3$, $NO_3$ for cineole, $\Delta^3$-carene, OMT, farnesene and OSQ. The related chemical reactions of stabilised Criegee intermediates (sCIs) with updated reaction rates from Boy et al. (2013) are also taken into account in current simulations. For more details about emissions and chemistry we refer to Mogensen et al. (2015)."

2. **Comments: (2) Due to the incomplete model description, it is not obvious for the reader that the vertical mixing of O3 is calculated similarly to that of any other compound in SOSAA (Eqs. 5 and 6), and that the "multi-layer O3 deposition model" actually consists of the few resistance terms shown in Fig. 1 (of which not all are effective). As SOSAA has previously been used for simulating the exchange of reactive compounds and latent heat within a forest canopy, obviously it must include some sort of description of the surface exchange processes corresponding to stomatal uptake at least. This relationship should be explained, especially for the stomatal resistance of both overstory and understory vegetation.**

Answer:

1. As previously indicated, we added more details about the turbulent mixing which clarifies how the vertical mixing is calculated. Furthurmore, we improved the prognostic equation for the evolution of the $O_3$ concentration for each layer and other compounds also follow this prognostic equation in SOSAA:

$$\frac{\partial [O_3]}{\partial t} = \frac{\partial}{\partial z}\left(K_t \frac{\partial [O_3]}{\partial z}\right) - V_d[O_3]A + Q_{chem} \tag{6}$$

where the first term on the right-hand side represents the vertical mixing of $O_3$. The second term is the sink by dry deposition which is non-zero only inside the canopy. The last one is chemistry production and loss for $O_3$ for each model layer. $V_d$ is the total dry deposition velocity at height z which already includes the uptake by the leaves, including the leaf stomata (see below), cuticle and the uptake by the soil for the understory layer. We also distinguish the difference in uptake by dry and wet leaves. $A$ is a unit scale factor which is set to 1 m$^2$ m$^{-3}$ here.

2. $r_{mes}$ can be neglected for $O_3$.

3. In SOSAA, the stomatal resistance for water vapor $r_{stm,H_2O}$ is computed by the SCADIS module. It is used to calculate the latent heat flux and thus the energy balance. The detailed

description of the formula refers to Sogachev et al. (2002). Then $r_{stm}$ for $O_3$ is obtained as

$$r_{stm} = \frac{D_{H_2O}}{D_{O_3}} r_{stm,H_2O} \tag{7}$$

Here $D_{H_2O}$ and $D_{O_3}$ are the molecular diffusivities of water vapor and $O_3$, respectively.

3. **Comments: (3) The presentation of input data should be clearer and specify the data actually used for the deposition calculations, i.e. which data are taken from measurements and what is derived within SOSAA. This is important, as a large part of the paper is dedicated to testing the modelled meteorological variables. For example, a comparison with observations is presented for u\*, but it is not explained how the modelled profile is obtained or how it is utilised in the model.**

Answer:

1. We added more details about SOSAA description (see above) which clarifies how variables are calculated in the model. We also improved the description of the input data for the model as (this paragraph is also shown in reply 1):

"The upper boundary values of $u$, $v$, $T$ and $q_v$ are constrained by the ERA-Interim reanalysis dataset provided by the European Centre for Medium-Range Weather Forecasts (ECMWF, Dee et al., 2011). At the canopy top, the incoming direct and diffuse global radiations measured at SMEAR II station, and the long wave radiation obtained from the ERA-Interim dataset are read in to improve the energy balance closure. Other radiation terms are computed according to the radiation scheme in Sogachev et al. (2002). At the lower boundary, the measured soil heat flux at SMEAR II are used to further improve the representation of surface energy balance. All the input data are interpolated to match the model time for each time step. With the input data, the mass and energy exchange between atmosphere and plant cover (including the soil underneath) and the radiation attenuation inside the canopy are optimal to simulate the micrometeorological drivers of $O_3$ deposition at this site."

2. $u_*$ is calculated in SCADIS for each layer with turbulent eddy diffusivity and the wind gradient. It can represent the shear stress and thus the turbulent strength. $u_*$ is also used to calculate the soil boundary layer resistance $r_{bs}$.

4. **Comments: (4.1) r_soil is modified from a default value based on model simulations, which are not discussed. These simulations should be shown and would serve as a useful sensitivity test.**

Answer:

1. Now we use the default value 400 s m$^{-1}$ proposed by Ganzeveld and Lelieveld (1995) and we add a soil boundary layer resistance $r_{bs}$.

2. We also did a sensitivity test for $r_{soil}$ with values of 200, 400, 600, 800 s m$^{-1}$. In general 400 s m$^{-1}$ resulted in a simulation of $O_3$ fluxes and in-canopy concentration profiles in best agreement with observations. The analysis is added in the revised manuscript as:

"$r_{soil}$ varied in different studies, ranging from 10 to 180 s m$^{-1}$ for dry soil and 180 to 1100 s m$^{-1}$ for wet soil (Massman, 2004). In this study the dry deposition module was developed on the basis of the model from Ganzeveld and Lelieveld (1995) in which $r_{soil}$ is 400 s m$^{-1}$. In order to assess the uncertainties involved in estimating $r_{soil}$, different values of $r_{soil}$ ranging from 200 to 800 s m$^{-1}$ were tested in this study (Table 1). As can be expected, the modelled $O_3$ fluxes decreased as $r_{soil}$ increased. The BASE case showed the best performance in general, although it overestimated $\sim 16\%$ nighttime $O_3$ fluxes. Since the RSOIL200 case overestimated $O_3$ fluxes by $\sim 17\%$ in average for the whole month, $\sim 12\%$ at daytime and $\sim 35\%$ at nighttime, the RSOIL200 sensitivity case indicates that using this lower estimate, a value that might be more appropriate for high organic (and dry) soils, seems to not properly represent the role of

Table 1: The average and standard deviation of modelled and measured $O_3$ fluxes above the canopy during different time periods (ALL for the whole month, D for daytime, N for nighttime) for different cases (OBS for measurement, BASE for basic settings used in this study, RSOIL200 uses the same settings as in BASE except $r_{soil} = 200$ s m$^{-1}$, similarly, RSOIL600 with $r_{soil} = 600$ s m$^{-1}$ and RSOIL800 with $r_{soil} = 800$ s m$^{-1}$) are shown. The relative error of modelled $O_3$ flux compared to the observation $(F_{t,mod} - F_{t,obs})/F_{t,obs}$ is also listed within the parentheses.

| cases | ALL | D | N |
|---|---|---|---|
| OBS | $0.125 \pm 0.090$ | $0.171 \pm 0.085$ | $0.052 \pm 0.037$ |
| RSOIL200 | $0.146 \pm 0.090$ (+16.6%) | $0.192 \pm 0.085$ (+12.3%) | $0.070 \pm 0.034$ (+34.9%) |
| BASE | $0.128 \pm 0.079$ (+1.93%) | $0.168 \pm 0.075$ (-1.51%) | $0.061 \pm 0.030$ (+16.1%) |
| RSOIL600 | $0.118 \pm 0.075$ (-5.85%) | $0.156 \pm 0.070$ (-8.64%) | $0.055 \pm 0.029$ (+5.07%) |
| RSOIL800 | $0.112 \pm 0.072$ (-10.7%) | $0.148 \pm 0.067$ (-13.0%) | $0.051 \pm 0.028$ (-2.28%) |

soil removal at this site. On the other hand, taking higher resistance values, e.g., one of 600 or 800 s m-1 seems to result in a better simulation of the role of the soil uptake at nighttime. However, considering the overall performance and better estimation of daytime $O_3$ fluxes, we still use 400 s m$^{-1}$ as the soil resistance."

5. **Comments: (4.2) r_ac is set to a very small arbitrary value. What is the point of including a resistance of 1 s m-1 in series with a resistance of 600 s m-1?**

Answer: In our model the role of turbulent transport, represented by the term $r_{ac}$, exists but is ignored for this particular layer. Because it is a very small term compared to the other processes (e.g., molecular diffusion and surface uptake).

6. **Comments: (4.3) r_b depends on molecular diffusivity and wind speed (or friction velocity, p.9). Please present the formula or an exact reference.**

Answer: The applied relationship is according to Meyers (1987). The reference will be included in the revised manuscript.

7. **Comments: (4.4) r_stm is calculated from evapotranspiration rate in SOSAA. How?**

Answer: It is described in Answer of Comments (2).

8. **Comments: (4.5) r_mes, r_cut and r_ws have constant values. Where do these come from?**

Answer: They are from Ganzeveld and Lelieveld (1995). We will also add the detailed information in the revised paper.

9. **Comments: (4.6) f_wet is a function of RH, for which the authors cite a grey literature report that does not even include original data on canopy wetness. Isn't there anything more substantial available?**

Answer: We will add another reference: Wu et al. (2003).

10. **Comments: (5) An ineffective aerodynamic resistance term (r_ac) is included in series with the soil resistance (cf. Comment 4.2 above). However, a much more important term, namely the near-soil boundary layer resistance, is ignored in the model.**

Answer: We will add a soil boundary layer resistance $r_{bs}$ in the revised manuscript as:

"The $r_{bs}$ is the soil boundary layer resistance which is calculated as (Nemitz et al., 2000),

$$r_{bs} = \frac{\mathrm{Sc} - \ln(\delta_0/z_*)}{\kappa u_{*g}} \tag{8}$$

Here Sc (1.07) is the Schmidt number for $O_3$. $\kappa$ is the von Kármán constant (0.41). $\delta_0 = D_{O_3}/(\kappa u_{*g})$ is the height above ground where the molecular diffusivity is equal to turbulent eddy diffusivity. $z_*$ (0.1 m) is the height under which the logarithmic wind profile is assumed. $u_{*g}$ is the friction velocity near the ground."

11. **Comments: (6) The leaf surface resistance has a general formulation (Eq. 3) so as to represent both needle-shaped and broad leaves. This is accomplished by a scaling factor (alpha = 0.5) introduced to account for one-sided stomatal exchange on leaves. As the non-stomatal exchange takes place on both sides of such a leaf, with separate boundary layers, this scaling does not result in a correct formula for deposition on two-sided leaves. The authors describe alpha as a correction factor, so it may represent an approximation. However, this approximation should be justified.**

Answer: We thank the reviewer for pointing out this flaw in the implementation of the dry deposition scheme in SOSAA. Now we modified the scheme as:

"$r_{veg}$ is the leaf surface resistance which represents how $O_3$ finally deposits onto different parts of leaf surface (Fig. 1). It can be calculated at each layer for needle leaves as

$$r_{veg} = r_b + \frac{1}{1/(r_{stm} + r_{mes}) + (1 - f_{wet})/r_{cut} + f_{wet}/r_{ws}} \tag{9}$$

While for broad leaves, $O_3$ can deposit on a side without stomata or a side with stomata, hence $r_{veg}$ is computed in a different way as

$$r_{veg} = 2 \left/ \left( \frac{1}{r_{veg1}} + \frac{1}{r_{veg2}} \right) \right. \tag{10}$$

$$r_{veg1} = r_b + \frac{1}{(1 - f_{wet})/r_{cut} + f_{wet}/r_{ws}} \tag{11}$$

$$r_{veg2} = r_b + \frac{1}{1/(r_{stm} + r_{mes}) + (1 - f_{wet})/r_{cut} + f_{wet}/r_{ws}} \tag{12}$$

Here $r_b$ is the quasi-laminar boundary layer resistance over the leaf surface, which depends on molecular diffusivity and horizontal wind speed (Meyers, 1987)."

We found that the overall impact of this flaw is small so the initially presented results are still valid. For example, for a typical condition at daytime ($r_{stm}$=2000.0 s m$^{-1}$, $f_{wet}$=0.0, $r_b$=150.0 s m$^{-1}$), the new $r_{veg}$ is about 3% larger than the old value for broad leaves.

12. **Comments: (7) The authors demonstrate that the model performs well in high humidity conditions but fails during the night-time when RH is low. As wet needle surfaces require additional parameterisations (RH dependent resistance, wet surface fraction), it is surprising that the authors do no first try to develop a parameterisation that performs well in dry conditions, i.e. in a much simpler case. Instead, they refer to simulation and measurement problems due to weak turbulence, but do not explain how these would depend on RH. If the measurement uncertainties increase with weakening turbulence and affect the model validation (and the u\* screening does not help), then this could be easily tested. As the soil resistance is given as a plausible explanation for the mismatch, it would also**

[Figure]

Figure 1:

**seem logical to test if a better fit can be obtained by varying this resistance (see Comment 4.1. above).**

Answer: This study also addresses the role of soil uptake but the reviewer is indeed correct that we didn't introduce a detailed representation of leaf/needle cuticle uptake as a function of RH. We agree with this but we also consider that there are only 69 data points under NL condition (nighttime with low humidity condition), which is a small portion compared to the total available data points of 886. This is also one reason of low correlation between the modelled and meaursed results for NL condition. Furthermore, the low humidity condition occurs less often at nighttime, the simulation bias for this condition only affects slightly the overall performance of the model. Therefore, we decide not yet to introduce such further modifications and rather focused on soil uptake. Further improvement on the wet skin fraction uptake will be focus of future studies with SOSAA.

13. **Comments: (8) Only ozone fluxes are considered in the analysis. As the modelled flux depends on the modelled concentration, which is affected by various processes and has a systematic diurnal cycle, it is difficult to assess how well the deposition processes are modelled by comparing fluxes alone. Perhaps you could have a look at the flux/concentration ratio (commonly called deposition velocity)?**

Answer: In this study we also compared the $O_3$ concentration profile in Fig. 8 which is also used to varify the model results. Therefore, the agreement between modelled and observed fluxes above the canopy and the concentration profiles inside the canopy we can conclude that the deposition processes are modelled quite well.

14. **Comments: (9) As the ozone fluxes measured at SMEAR II have been analysed in a large number of previous studies, including two different multi-layer models (Rannik et al., 2012; Launiainen et al., 2013), I would expect to see a more systematic comparison of these results. From the process modelling point of view, it would be useful to discuss how and why the modelled results differ between the three multi-layer models.**

Answer: Our study revealed some differences between our model and these two previous models, and showed the novel points of our current model by inclusion of the following statement (also see the reply below):

"Two different studies that also applied multi-layer models (Rannik et al., 2012; Launiainen et al., 2013) to simulate the $O_3$ fluxes and concentration inside the boreal forest canopy had their limitations on estimating the chemical contribution. Rannik et al. (2012) only considered one chemical reaction of $O_3$ with $\beta$-caryophyllene. In Launiainen et al. (2013), they simplified the chemical production and loss of $O_3$ with only two parameters to represent the first-order

kinetic sink and photo-chemical production. In this study, we implemented a chemistry module with a detailed list of chemical reactions, which was able to provide a more accurate estimation of chemical removal of $O_3$ inside the canopy."

15. **Comments: (10) The discussion of chemical removal (Sect. 3.7) is based on the reactivity estimates obtained from the literature. According to the model description, the SOSAA model employed here includes a detailed chemistry module, which I assume was used in the present simulations. Why are these calculations not utilised for estimating the importance of in-canopy chemistry?**

Answer: In the revised manuscript we included the role of in-canopy chemical transformations on $O_3$ deposition by using the chemical module. In this way, we calculated the diurnal cycle of the net effect of chemical processes which are able to destroy $O_3$ by reacting with other compounds or produce $O_3$ by photochemical reactions. The analysis is as follows:

"In order to get rid of the effect of synoptic-scale transport of $O_3$ and only focus on the local sinks and sources, we implemented the case FREEO3. In this simulation case we ignored the role of advection and only considered the role of local sources and sinks inside the canopy, i.e., dry deposition, chemical production and loss, and turbulent transport. Here the time period from Aug. 5th to 14th were selected from the simulation results to analyze the local chemical contribution, because the modelled $O_3$ concentration fitted to the measurement the best during this period out of the whole month for the case FREEO3, which indicated that the advection only had little effect on the local observed $O_3$ variation. The daily averaged (from Aug. 5th to 14th) production and loss of $O_3$ inside the canopy caused by dry deposition ($F_{depo}$) and chemistry ($F_{chem}$) are plotted in Fig. 2. The unit nmol m$^2$ s$^{-1}$ means that how much nmol $O_3$ inside the canopy alters per unit square meter per second. So positive values correspond to $O_3$ production and negative values represent $O_3$ loss. Here the chemistry production is a net effect of $O_3$ loss reactions and photo-chemical production. $F_{depo}$ (obviously negative) shows a maximum $O_3$ loss rate at about 14:00. While the chemistry produces $O_3$ from morning at $\sim$ 06:00 to the afternoon at $\sim$ 15:00, and destroys it throughout the other moments of the day, especially at nighttime (Fig. 2). The ratio between $F_{chem}$ and $F_{depo}$ shows that chemical removal has its largest contribution of $\sim$ 9% of the dry deposition sink in average at nighttime from 20:00 to 04:00. At daytime, our model simulations indicate that the $O_3$ production caused by chemistry can compensate up to $\sim$ 4% of dry deposition loss in average. However, during the selected period, the chemical contribution and compensation can reach up to $\sim$ 24% and $\sim$ 20% at most. This indicates that in general chemistry has minor impact on $O_3$ alteration, but at some specific time the chemical production and removal of $O_3$ can still play a significant role."

16. **Comments: (11) Even though I indicated in my access review that a linguistic revision is necessary, there are still numerous errors, some of which impair presentation. A few examples are given in the detailed comments below.**

17. **Comments: P1/L23: "under current knowledge of air chemistry" is obvious so can be removed.**

After adding the chemistry part in our simulation, the whole paragraph:

"Furthermore, a qualitative evaluation of the chemical removal time scales indicated that the chemical removal rate within canopy was about 5% of the total deposition flux at daytime and 16% at nighttime under current knowledge of air chemistry."

has been rewritten as:

"The chemical contribution to $O_3$ removal has been evaluated directly in the model simulations. According to the simulated averaged diurnal cycle the net chemical production of $O_3$ compensates up to $\sim$ 4% of dry depositon loss from $\sim$ 06:00 to $\sim$ 15:00. During nighttime, the net chemical removal of $O_3$ further enhanced removal by dry deposition by a maximum $\sim$

[Figure]

Figure 2: (a) The daily averaged (from Aug. 5th to 14th) production and loss caused by chemistry ($F_{chem}$, red) and dry deposition ($F_{depo}$, blue). (b) The ratio between $F_{chem}$ and $F_{depo}$. Zero lines for $F_{chem}$ and the ratio are plotted as dashed lines. Shaded areas show the range of $\pm 1$ standard deviation.

9%. This indicates that there appears to be an overall relative small contribution by airborne chemical processes on $O_3$ removal at SMEAR II station."

18. **Comments: P2/L24: Any more recent studies?**

We have not found very recent review papers focusing on the $O_3$ uptake on the wetness leaf surface, so we changed the statement as:

"Among them the effect of canopy wetness on $O_3$ deposition has attracted a lot of attention in previous studies (e.g., Massman, 2004; Altimir et al., 2006)."

19. **Comments: P2/L31-: A reference is needed for "the boreal forest emits a large portion of BVOCs". The examples discussed are for California.**

"the boreal forest emits a large portion of BVOCs"
changed to
"the boreal forest emits a large portion of BVOCs (Rinne et al., 2009)".

20. **Comments: P3/L9-12: Unclear logic. It is not only removal processes that are relevant. The introduction of eddy-covariance measurements to the discussion seems awkward. Please reformulate.**

"These removal processes altogether determine the contribution of $O_3$ uptake on forest ground surface and understory vegetation, the vertical distribution of $O_3$ concentration as well as the non-stomatal uptake contribution, which are considered as three crucial challenges to understand the relationship between the eddy-covariance measurements and $O_3$ uptake (Launiainen et al., 2013). Therefore several numerical models ..."
changed to
"Last two decades, several numerical models ..."

21. **Comments: P3/L14: Wesely (1989) describes a single model, which is based on the big-leaf approach. So this sentence ("Among these models ...") makes little sense.**

"... different climatic and environmental conditions, which are generally based on the surface deposition model described by Wesely (1989). Among these models, the so-called "big-leaf" approach method is widely used and usually coupled to ..."

changed to

"... different climatic and environmental conditions. Many of them have implemented the big-leaf framework following the Wesely (1989) approach which can be coupled to ..."

22. **Comments: P3/L14-15, "usually": Very often big-leaf models are used as inferential models.**

"usually" changed to "can be".

23. **Comments: P3/L15: Zhang et al. (2002) deal with deposition parameterisations rather than large-scale modelling.**

"e.g. Zhang et al., 2002" changed to "e.g., Hardacre et al., 2015".

24. **Comments: P3/L18: Altimir et al. (2006) do not employ a multi-layer model.**

"e.g. Ganzeveld et al., 2002b; Altimir et al., 2006; Rannik et al., 2012; Launiainen et al., 2013" changed to

"e.g. Ganzeveld et al., 2002b; Rannik et al., 2012; Launiainen et al., 2013"

25. **Comments: P3/L23: A paper from 2000 is hardly suitable for evaluating recent models.**

We removed this sentence since it is not closely relevant to current discussion:

"Recent models have been developed more and more based on the physical, chemical and biological processes under actual environmental conditions, which reduce the dependency of empirical parameters (Wesely and Hicks, 2000)."

26. **Comments: P3/L24, "process-based": Unclear which processes are referred to here. The implementation consists of a largely empirical resistance parameterisation.**

"a multi-layer process-based $O_3$" changed to "a multi-layer $O_3$ ...".

27. **Comments: P3/L33: Unclear which challenges are referred to here.**

"... for validating the new model and also shining a light on those three challenges with the model."
changed to
"... for validating the new model and investigating more detailed processes."

28. **Comments: P4/L23: What is meant by "the same below"?**

It means all the height levels mentioned below are referring to above the ground level. I modified the sentence here.

"... 67.2 m (above the ground level, the same below), ..." changed to "... 67.2 m above the ground level, ...".

29. **Comments: P4/L27-28: Why was the ozone flux calculated with data from a different anemometer than for other fluxes?**

The sensible and latent heat flux measurements were performed at a tower located at about 25 m distance from the $O_3$ flux measurement tower. Hence a different anemometer was used to obtain the $O_3$ fluxes.

30. **Comments: P4/L31: Did you correct the O3 flux data for high-frequency losses? How large were the corrections?**

Yes, sure. At this site Keronen et al. (2003) reported the correction factors 1.03–1.19 for unstable and 1.13–1.22 for stable stratification conditions (Figs. 3 and 4 in Keronen et al., 2003).

31. **Comments: P5/L28: What does "partly constrained" mean?**

    "partly constrained by" changed to "constrained by".

32. **Comments: P6/L11: I would recommend against using the term "deposition velocity" for layer-specific conductances.**

    We will change every layer-specific "deposition velocity" to "layer-specific conductance".

33. **Comments: P6/L19-20: What does "the unit is the same ..." mean?**

    It means all the resistance shown below have the same unit "s m$^{-1}$". Now we removed this "the unit is the same ..." and reorganized the introduction of the resistance scheme.

34. **Comments: P6/L27-28: Unclear language; please rephrase.**

    We reorganized this resistance scheme as shown in reply 11.

35. **Comments: P7/ Eq.5: This is a strange combination of partial derivatives and finite differences. Please present the equation in a mathematically consistent form. You also need to assume constant air density here. It would be more appropriate to present the 'flux' as mass flux density (g m-2 s-1).**

    Answer: 1. The air density is constant in our model and we modified the prognostic equation for the simulated changes in $O_3$ concentration as mentioned above (reply 2):

    $$\frac{\partial[O_3]}{\partial t} = \frac{\partial}{\partial z}\left(K_t \frac{\partial[O_3]}{\partial z}\right) - V_d[O_3]A + Q_{chem} \tag{13}$$

    where the first term on the right-hand side represents the vertical mixing of $O_3$, the second term is dry deposition sink and the last one is chemistry production and loss for $O_3$. $V_d$ is the total deposition velocity at height z including the vegetation and soil uptake. $A$ is a unit scale factor which is set to 1 m$^2$ m$^{-3}$ here.

    2. This is a good suggestion. We will change the unit of flux to nmol m$^{-2}$ s$^{-1}$ or ng m$^{-2}$ s$^{-1}$, and correspondingly change $O_3$ concentration unit to nmol m$^{-3}$ or ng m$^{-3}$.

36. **Comments: P7/Eq.5: How did you solve for [O3]. If it is a common procedure within SOSAA, perhaps you could explain it in Sect. 2.3.1.**

    We explained more details in the section of SOSAA model, including a more detailed description of the calculation of turbulent mixing. All the other compounds are computed in the same way as $O_3$ shown here (see reply 2).

37. **Comments: P8/L1-2: How did you do the forcing? Fig. 2b does not explain this.**

    We forced the $O_3$ concentration at 23 m to resemble the observed value every time step, the $O_3$ concentration at other levels are then calculated by Eq. 6. In this way, we implicitly added the role of advection in determining the surface layer (23 m) $O_3$ concentrations. Fig. 2b shows the gap-filled observed values which are used for the forcing.

38. **Comments: P9/Table 1: u* is not limited to the canopy top.**

    "friction velocity at the canopy top" changed to "friction velocity".

39. **Comments: P9/L13, P10/L1, P11/L7, "was calculated": How? These should be moved to the methods description.**

    1. We added this sentence in SOSAA description (also see reply 1):

    "Then the reflection, absorption, penetration and emission of three bands of radiation (long-wave, near-infrared and PAR) at each layer inside the canopy are explicitly computed according to the radiation scheme proposed by Sogachev et al. (2002)."

2. "The PAR on top of the canopy was calculated directly from the input incoming short wave radiation with a daytime maximum of about 250–300 W m$^{-2}$ during the simulation month. Inside the canopy, PAR was calculated by considering the absorption, reflection and scattering effects of canopy leaves (Sogachev et al., 2002)."
changed to
"The PAR on top of the canopy was calculated directly from the measured incoming short wave radiation serving as input to the model, whereas PAR inside the canopy was calculated by considering the absorption, reflection and scattering effects of canopy leaves (Sogachev et al., 2002)."

3. Removed "The simulated O3 turbulent flux was calculated from the O3 concentration gradient and the turbulent eddy diffusivity at 23 m." since it is explained already in the model description part.

40. **Comments: P10/Fig. 2b: Gap-filling of data is not described in the paper. Why was it performed? For which variables?**

Sometimes the instruments do not work or the quantities are lower than the detection limit, so we need to fill the gaps then use them as the input for the model. We will add a sentence to clarify this:

"The missing observed data points of T, RH and $O_3$ were gap-filled with the method described in Gierens et al. (2014)."

41. **Comments: P10/L8 (also elsewhere): These data are measured well above the canopy, so why are they referred to as "canopy top".**

We think 23 m is just above the canopy and can be considered as the canopy top. We will remove these texts in section titles and change "canopy top" to "above the canopy" if necessary.

42. **Comments: P11/Fig. 3, P13/Fig. 5: Are these data screened for low turbulence?**

No, they included the data from the whole month, including those days with low turbulence.

43. **Comments: P12/L11-12: How do the low humidity conditions affect turbulent mixing, making this difficult to simulate?**

Usually at nighttime RH is larger than 70% (NH condition), under this condition, the wet skin uptake contributes more than 50% to the deposition flux, so the turbulent mixing above the ground which affects the deposition flux onto soil only plays a minor role on the deposition flux above the canopy. However, in NL condition which does not happen frequently, nearly all the deposition inside the canopy is caused by soil deposition. Hence, the difficulty of simulating the exchange processes near the surface may cause more difficulty of simulating the deposition flux into soil surface under NL condition than NH condition. Moreover, the impact of vertical advection of $O_3$ could be more significant in NL condition, which also complicates the analysis.

44. **Comments: P13/Table 2, P15/Fig. 7: Why is the R2 of the full data set higher than the R2 of any of the four subsets?**

This is due to the fact that the night-time observations are located close to zero, whereas daytime observations have larger absolute values but are relatively scattered. When combined, the nighttime observation improves the correlation statistics value by extending the daytime observation to zero, defining better linear relationship with improved $R^2$ value.

45. **Comments: P14/L6: Can you estimate how much the correlation was affected by random uncertainty?**

For the $O_3$ turbulent flux measurement at the same site Keronen et al. (2003) presented the random error statistics, defined as one standard deviation of the random uncertainty of

turbulent flux, ranging from about 10 to 40%. Such uncertainty contributed to the data scattering when comparing the modelled and measured fluxes, such as in Fig. 7, and reduced the correlation statistics. By assuming the most frequent flux relative random uncertainty value of 20%, we estimated numerically that the $R^2$ value is reduced by about 0.1 due to random uncertainty of flux errors. This is a rough estimate as the value depends on the distributions of the fluxes as well as their uncertainties, which are not exactly known for both measured and modelled estimates.

46. **Comments: P15/Fig. 7: The caption is difficult to read.**

That caption has been changed to:
"Scatter plots of modelled versus measured $O_3$ turbulent fluxes above the canopy. The data points are plotted separately for different groups (DH, DL, NH and NL) with their $R^2$ values shown in the legend. $R^2$ of the whole dataset is shown below the legend."

47. **Comments: P16/L11: The notation related to the cumulative flux is not obvious.**

The statement on the cumulative flux calculation has been changed to:
"The normalized cumulative $O_3$ deposition flux at layer $i$ can be obtained as

$$F_{c,i} = \frac{\sum_{k=1}^{i} F_k}{\sum_{k=1}^{N} F_k} \tag{14}$$

where $F_k$ is the $O_3$ deposition flux at layer $k$ and N is the layer index just above the canopy. The profiles of $F_c$ and the contributions of different deposition pathways for four different conditions were shown in Fig. 9."

48. **Comments: P16/L14: No stomatal contribution is indicated for the understory vegetation in Figure 9.**

There is $\sim 5\%$ deposition flux from stomatal uptake by the understory vegetaion at daytime (Fig. 9). So we used "little contribution".

49. **Comments: P16/L14-P17/L4: Unclear presentation. Does "uptake on leaf surfaces" refer to the flux or the cumulative flux (accumulated from the bottom)?**

"uptake on leaf surfaces" changed to "cumulative uptake on leaf surfaces".

"in the NL condition when both the stomatal uptake and wet skin uptake were limited." changed to "in the NL condition when both the cumulative stomatal uptake and wet skin uptake were limited."

50. **Comments: P17/Fig. 9: What explains the stomatal uptake during the nighttime?**

Caird et al. (2007) showed that the stomata are not completely closed at night and several sources might affect the nocturnal stomatal conductance of water vapor, e.g., vapor pressure deficit, water availability (Caird et al., 2007). In SOSAA, a high value of about 13800 s m$^{-1}$ is used for nighttime stomatal resistance of water vapor.

51. **Comments: P17/L11: Please quantify the "limited O3 uptake", as it is obvious that small surface area corresponds to small uptake.**

"providing limited $O_3$ uptake compared to the total $O_3$ deposition." changed to "providing less than 2% $O_3$ uptake compared to the total $O_3$ deposition."

52. **Comments: P17/L12-13: These percentage contributions only refer to the mean values of the four data sets, so discussion of variation may be misleading here.**

"As a result, the simulated non-stomatal contribution to the integrated $O_3$ deposition flux above the canopy varied from 33–56% during daytime to 85–92% during nighttime (Table 3)."

changed to
"As a result, the simulated averaged non-stomatal contribution to the integrated $O_3$ deposition flux above the canopy was 37% during daytime and 96% during nighttime (Table 3)."

Here we made some modifications to the resistance scheme according to the comments, so the values here are not the same as the original manuscript.

53. **Comments: P17/P13: This may be explained by Launiainen et al. (2013), but the meaning of the "sub-canopy layer" is unclear. Does it include some other vegetation surfaces in addition to the understory vegetation and soil?**

Here we used the same word "sub-canopy layer" as in Launiainen et al. (2013) to make comparison. The measurement height is 4.2 m in their research, so the sub-canopy layer here contains the understory vegetation and the soil surface below 4.2 m. No other additional vegetation is considered.

54. **Comments: P17/L14-17: The contributions cited from Launiainen et al. (2013) do not add up to 100%; why?**

35–45% is the sub-canopy layer contribution to the total $O_3$ deposition flux at daytime, and 25–30% is the sub-canopy layer contribution at nighttime, so they refer to the same quantity at different time periods. Therefore, they do not add up to 100%.

55. **Comments: P18/L1: How was the soil resistance determined in the first place?**

According to Ganzeveld and Lelieveld (1995) $r_{soil}$ is 400 s m$^{-1}$, we also did a sensitivity test for $r_{soil}$ and found that in general applying this global mean estimate of $r_{soil}$ as 400 s m$^{-1}$ appeared to result in the best simulation of $O_3$ deposition fluxes and in-canopy concentrations at this site.

56. **Comments: P18/L3-5: I do not see how this conclusion about EC measurements results from the data presented here.**

The reviewer is correct, so this statement has been changed to
"Therefore, we expected that the poor performance for the NL condition also resulted from the limited data points under this condition (only 69 data points) which leads to larger ratio of random uncertainty and thus smaller $R^2$."

57. **Comments: P19/L8-9: You should explain how these percentages were obtained.**

This statement has been changed to
"These estimates showed that the chemical removal accounted for about 5% ($3384/63291 \approx 5\%$) and 16% ($9349/59880 \approx 16\%$) of the total $O_3$ removal within the canopy at daytime and nighttime, respectively."

58. **Comments: P19/L24: No data on BVOC removal are presented in this study.**

We are preparing a document on the role of canopy deposition in BVOC exchange for this site. However, the reviewer is right in that we do not further present here any results on BVOC deposition and consequently the statement has been changed to
"... e.g., by the dry and wet cuticle, by stomatal uptake and by the soil surface."

59. **Comments: P20/L14: Poor presentation; please rephrase.**

The statement has been changed to
"Our study indicates that uptake by the wet canopy appears to dominate nocturnal removal at this site with a relative smaller role of soil removal especially during high humidity conditions."

60. **Comments: P20/L19-20: I do not see how these different flux partitionings would indicate "the difficulty of simulating and measuring O3 deposition at night".**

Removed "This also indicated the difficulty of simulating and measuring $O_3$ deposition at night with weak turbulence (Rannik et al., 2009)."

61. **Comments: Technical comments**

62. **Comments: P1/L18,L19: Incorrect grammar.**

"was similar to" changed to "was similar as the contribution reported in"

"two times as" changed to "two times larger than"

63. **Comments: P2/L1-3: Unnecessary material for the abstract.**

Removed "The evaluation of the $O_3$ deposition processes provides improved understanding about the mechanisms involved in the removal of $O_3$ for this boreal forest site which are also relevant to the removal of other reactive compounds such as the BVOCs and their oxidation products, which will be focus of a follow-up study."

64. **Comments: P3/L27-28: "manuscript in preparation" is not a useful reference.**

"(MLC-CHEM, manuscript in preparation)." changed to "(MLC-CHEM, e.g., Ganzeveld et al. (2002))"

65. **Comments: P5/L10: Incorrect grammar.**

"a more strictly criteria" changed to "a more strict criteria"

66. **Comments: P8/L8: Incorrect grammar.**

"The time series of temperature especially this transition were well predicted by the model (Fig. 2a)."
changed to
"Analysis of the full temperature record indicates that this transition in the weather conditions at the site was well simulated by the model."

67. **Comments: P8/L13: Repetition from the introduction.**

Removed "It was also interesting to study this featured time period with hot and dry climate which probably represented a future trend at this boreal forest site (Williams et al., 2011)."

68. **Comments: P9/L1 (also elsewhere): replace "showed" by "shows".**

"Figure 3 showed the comparison results ..." changed to "Figure 3 shows the comparison results ..."

"Figure 3a showed the good agreement ..." changed to "Figure 3a shows the good agreement ..."

"Figure 7 showed the correlation ..." changed to "Figure 7 shows the correlation ..."

69. **Comments: P12/L9,L13: Incorrect grammar.**

L9: "followed by the condition DH with $R^2$ of 0.30, both of them were under high humidity conditions."
changed to
"followed by the results reflecting the daytime high humidity conditions. Note that these conditions with highest correlations were also the conditions with high relative humidity."

L13: "the nighttime $O_3$ turbulent flux were affected by" changed to "the nighttime $O_3$ turbulent flux was affected by"

70. **Comments: P17/L17: Typo.**

   "Tabel 3" changed to "Table 3"

71. **Comments: P20/L9: Incorrect grammar**

   "were significant in the total $O_3$ uptake" changed to "were significant for the total $O_3$ uptake"

**References**

[revised manuscript text omitted]

---

## Author Comment (AC2) · 6 Oct 2016

**Reply to comments on "Simulating ozone dry deposition at a boreal forest with a multi-layer canopy deposition model"**

October 6, 2016

**We thank the reviewer's thoughtful comments which are helpful not only for this manuscript but also for our future research. Our reply for all the comments are shown below.**

1. **Comments: 1. However, I would have appreciated a more extended parameterization and a better description of the model in order to clearly understand the formalism adopted to predict energy balance terms.**

   We added more details about the energy balance terms, including sensible and latent heat fluxes, soil heat flux and radiation.

   "In SOSAA, the horizontal wind velocity ($u$ and $v$), temperature ($T$), specific humidity ($q_v$), turbulent kinetic energy (TKE) and the specific dissipation of TKE ($\omega$) are computed every time step (10 s) by prognostic equations. In order to represent the local to synoptic scale effects, $u$, $v$, $T$ and $q_v$ near and within the canopy are nudged to local measurement data at SMEAR II station with a nudging factor of 0.01. A TKE-$\omega$ parameterization scheme is used to calculate the turbulent diffusion coefficients ($K_t$) (Sogachev, 2009),

   $$K_t = C_\mu \frac{\text{TKE}}{\omega} \tag{1}$$

   $$\omega = \frac{\varepsilon}{\text{TKE}} \tag{2}$$

   where $\varepsilon$ is the dissipation rate of TKE and $C_\mu$ is a closure constant. Hence the turbelent flux of a quantity $X$ ($F_{t,X}$) can be computed as

   $$F_{t,X} = -K_t \frac{\partial X}{\partial z} \tag{3}$$

   where upward fluxes are positive and vice versa. Specifically, the sensible heat flux (H) and latent heat flux (LE) at each model layer are computed as

   $$\text{H} = -C_{p,air}\rho_{air}K_t\left(\frac{\partial T}{\partial z} + \gamma_d\right) \tag{4}$$

   $$\text{LE} = -L_v K_t \frac{\partial q_v}{\partial z} \tag{5}$$

   where $C_{p,air}$ (1009.0 J kg$^{-1}$ K$^{-1}$) is the specific heat capacity at constant pressure. $\rho_{air}$ (1.205 kg m$^{-3}$) is the air density which is a constant in the model. $\gamma_d$ (0.0098 K m$^{-1}$) is the lapse rate of dry air. $L_v$ (2.256 × 10$^6$ J kg$^{-1}$) is the latent heat of vaporization for water."

   "The upper boundary values of $u$, $v$, $T$ and $q_v$ are constrained by the ERA-Interim reanalysis dataset provided by the European Centre for Medium-Range Weather Forecasts (ECMWF, Dee et al., 2011). At the canopy top, the incoming direct and diffuse global radiations measured

Table 1: The average and standard deviation of modelled and measured (OBS) $O_3$ fluxes above the canopy for different conditions in different cases are shown. The relative change of modelled $O_3$ flux compared to the observation $(F_{t,mod} - F_{t,obs})/F_{t,obs}$ is also listed within the parentheses.

| cases | ALL | D | N |
|---|---|---|---|
| OBS | $0.125 \pm 0.090$ | $0.171 \pm 0.085$ | $0.052 \pm 0.037$ |
| RSOIL200 | $0.146 \pm 0.090$ (+16.6%) | $0.192 \pm 0.085$ (+12.3%) | $0.070 \pm 0.034$ (+34.9%) |
| BASE | $0.128 \pm 0.079$ (+1.93%) | $0.168 \pm 0.075$ (-1.51%) | $0.061 \pm 0.030$ (+16.1%) |
| RSOIL600 | $0.118 \pm 0.075$ (-5.85%) | $0.156 \pm 0.070$ (-8.64%) | $0.055 \pm 0.029$ (+5.07%) |
| RSOIL800 | $0.112 \pm 0.072$ (-10.7%) | $0.148 \pm 0.067$ (-13.0%) | $0.051 \pm 0.028$ (-2.28%) |

at SMEAR II station, and the long wave radiation obtained from the ERA-Interim dataset are read in to improve the energy balance closure. Then the reflection, absorption, penetration and emission of three bands of radiation (long-wave, near-infrared and PAR) at each layer inside the canopy are explicitly computed according to the radiation scheme proposed by Sogachev et al. (2002). At the lower boundary, the measured soil heat flux at SMEAR II are used to further improve the representation of surface energy balance. All the input data are interpolated to match the model time for each time step. With the input data, the mass and energy exchange between atmosphere and plant cover (including the soil underneath) and the radiation attenuation inside the canopy are optimal to simulate the micrometeorological drivers of $O_3$ deposition at this site."

2. **Comments: 2. There are some arbitrary choices of parameters, and not a convincing analysis of sensitivity or results from a model calibration. A table showing results from a sensitivity analysis should be provided.**

We added a sensitivity test of $r_{soil}$ as below:

"$r_{soil}$ varied in different studies, ranging from 10 to 180 s m$^{-1}$ for dry soil and 180 to 1100 s m$^{-1}$ for wet soil (Massman, 2004). In this study the dry deposition module was developed on the basis of the model from Ganzeveld and Lelieveld (1995) in which $r_{soil}$ is 400 s m$^{-1}$. In order to assess the uncertainties involved in estimating $r_{soil}$, different values of $r_{soil}$ ranging from 200 to 800 s m$^{-1}$ were tested in this study (Table 1). As can be expected, the modelled $O_3$ fluxes decreased as $r_{soil}$ increased. The BASE case showed the best performance in general, although it overestimated $\sim 16\%$ nighttime $O_3$ fluxes. Since the RSOIL200 case overestimated $O_3$ fluxes by $\sim 17\%$ in average for the whole month, $\sim 12\%$ at daytime and $\sim 35\%$ at nighttime, the RSOIL200 sensitivity case indicates that using this lower estimate, a value that might be more appropriate for high organic (and dry) soils, seems to not properly represent the role of soil removal at this site. On the other hand, taking higher resistance values, e.g., one of 600 or 800 s m-1 seems to result in a better simulation of the role of the soil uptake at nighttime. However, considering the overall performance and better estimation of daytime $O_3$ fluxes, we still use 400 s m$^{-1}$ as the soil resistance."

3. **Comments: 3. Basic questions like: what could be the effect of an increase in air temperature and precipitation regimes on ozone deposition? Are not resolved, although it would have been nice triggering the model for some predictions of Ozone deposition under future environmental changes. In general the paper lacks of more mechanistic explanations of the results, with more discussion on the possible drivers of dry and wet ozone deposition.**

The reviewer has a point also since we have indicated that the observational dataset included data that were potentially resembling more common future conditions at this boreal forest site. However, in the present study we decided to limit ourselves to analyse the model performance

for the contrasting day and night time, wet and dry conditions to evaluate the role of the various substrates in the overall $O_3$ removal. This also reveals the potential significance of non-stomatal removal mechanisms at this site which calls for a better representation of these processes. Such a further improved model could then be applied in follow-up studies to assess what future climate change conditions could imply for removal of pollutants such as $O_3$ but also other related compounds over boreal forests.

4. **Comments: 4. Pag 2 line 25: You mention again that dew on leaves can increase deposition, but could you spend two lines mensioning the reasons or hypothesis why a hydrofobic molecul reacts so fast on wet surfaces?**

   We added this description in the introduction:

   "Previous studies showed that both the micro structure of the leaf surface and the hydrophilic compounds existing on the leaf surface are able to facilitate the formation of the water films or clusters, although the foliage surface itself is hydrophobic (Altimir et al., 2006). As a result, the different dissolved compounds like organics in the solution formed on leaf surface could react with $O_3$ and thus enhance the $O_3$ uptake (Altimir et al., 2006)."

5. **Comments: 5. Pag 3 line 10. What about NOx emitted from soils? Couldn't fast reactions between O3 and NO lead to high O3 fluxes in the sub-canopy region?**

   In SMEAR II station, NO emission is about 6 ng(N) m$^{-2}$ h$^{-1}$ which is close to the detection limit (Pilegaard et al., 2006). Moreover, according to the results in Rannik et al. (2009), the $O_3$ uptake due to reaction with NO emission is only about 0.0025% (10$^{-4}$ nmol m$^{-2}$ s$^{-1}$ / 4 nmol m$^{-2}$ s$^{-1}$) of the total nighttime $O_3$ flux. The sub-canopy $O_3$ flux at nighttime was about 25–30% of total $O_3$ uptake, so the effect of reaction with NO on sub-canopy $O_3$ flux can be ignored.

6. **Comments: 6. Pag 3 line 34: Only one month to test the model? The relative contributions of O3 sinks changes a lot during the seasons in repsonse to air temperature and plant phenology. It is a pity that such an important modelling effort is limited to one month, I would extend to the all vegetative season.**

   It would indeed be nice to conduct an analysis of a full seasonal cycle but this month was giving access to a complete dataset giving the best constraints for the presented detailed evaluation of the model also having still quite some large contrasts. Moreover, first assessing a proper representation of the main drivers of $O_3$ exchange would then also allow use of the model for full seasonal cycle studies in future research.

7. **Comments: 7. Pag 5 line 5: Extensive research has been conducted in Yuttiala to refine turbulence limitation to flux measurements. Why should we expect an ustar threshold different from other scalars measured at the site?**

   Different scalars may be differently affected by the nighttime phenomena such as accumulation, vertical as well as horizontal advection and in more general by stability conditions. This is due to build up of the concentration gradient which is expected to be particularly large for emitted compounds such as carbon dioxide. Ozone is instead deposited and therefore no large concentration gradients can form, meaning also that the mass balance components other than vertical transport are expected to be smaller. We use the criterion velocity threshold well justified for $O_3$ e.g. by Rannik et al. (2009).

8. **Comments: 8. Pag 6 line 20: do you have experience of subcanopy O3 fluxes so that you can better parameterize soil reisstances? It seems here that usage of one value rather than another is arbitrary and not properly calibrated.**

   The process of $O_3$ uptake by soil includes understorey transport ($r_{ac}$), diffusion at the soil/litter layer interface ($r_{bs}$) and, finally uptake by this soil/litter layer ($r_{soil}$) which might be strongly

affected by wetness. In this study we ignored $r_{ac}$ since the height of the lowest layer is only about 0.3 m above the ground where vertical transport is mainly limited by the molecular diffusion above the surface which is represented by $r_{bs}$. $r_{bs}$ will be added in the revised manuscript as:

"The $r_{bs}$ is the soil boundary layer resistance which is calculated as (Nemitz et al., 2000),

$$r_{bs} = \frac{\text{Sc} - \ln(\delta_0/z_*)}{\kappa u_{*g}} \tag{6}$$

Here Sc (1.07) is the Schmidt number for $O_3$. $\kappa$ is the von Kármán constant (0.41). $\delta_0 = D_{O_3}/(\kappa u_{*g})$ is the height above ground where the molecular diffusivity is equal to turbulent eddy diffusivity. $z_*$ (0.1 m) is the height under which the logarithmic wind profile is assumed. $u_{*g}$ is the friction velocity near the ground."

For the soil/litter layer resistance $r_{soil}$, we are aware that application of the value 400 s m$^{-1}$ deemed to represent the global mean soil uptake effciency and is thus a very crude simplication. However, from the conducted sensitivity analysis it can be inferred that this crude representation appears to result in the best representation of both $O_3$ deposition fluxes as well as $O_3$ concentration profiles inside the canopy. Actual confirmation of the correctness of the selected value can only be done conducting more detailed soil uptake measurements. Our study also clearly demonstrates the need for such additional measurements.

9. **Comments: 9. Pag 7 line 15. So you mean that Kt has been estimated form measured fluxes? Or in which other way? Reading through the manuscript I feel like the description of the model is not accurate, and more informations should be provided.**

   We added more detailed description about the model SOSAA as described above. So $K_t$ is calculated in the model from a TKE-$\omega$ scheme.

10. **Comments: 10. Pag 19 line 15: Can you say that NOx are also not relevant in the boreal forest?**

    Yes, from previous studies, we can conclude that NOx is not relevant to the $O_3$ uptake in SMEAR II station as we discussed above: At the SMEAR II station, NO emission is close to the detection limit (Pilegaard et al., 2006) and the $O_3$ uptake due to reaction with NO can be ignored (Rannik et al., 2009).

11. **Comments: 11. Pag 20 line 11: Since the Stomatal resistance is calculated based on evapotranspiration, are you sure that relevant nocturnal soil evaporation does not contribute significantly to Rc? Have you tried to separate canopy transpiration form soil evaporation in the model?**

    Actaully, the stomatal resistance is calculated based on the evapotranspiration from leaves and is already separated from soil evaporation in the model. Therefore, the soil evaporation does not contribute to stomatal conductance in the model.

**References**

Altimir, N., Kolari, P., Tuovinen, J.-P., Vesala, T., Bäck, J., Suni, T., Kulmala, M., and Hari, P. (2006). Foliage surface ozone deposition: a role for surface moisture? *Biogeosciences*, 3:209–228.

Dee, D. P., Uppala, S. M., Simmons, A. J., Berrisford, P., Poli, P., Kobayashi, S., Andrae, U., Balmaseda, M. A., Balsamo, G., Bauer, P., Bechtold, P., Beljaars, A. C. M., van de Berg, L., Bidlot, J., Bormann, N., Delsol, C., Dragani, R., Fuentes, M., Geer, A. J., Haimberger, L., Healy, S. B., Hersbach, H., Hólm, E. V., Isaksen, L., Kållberg, P., Köhler, M., Matricardi, M.,

McNally, A. P., Monge-Sanz, B. M., Morcrette, J.-J., Park, B.-K., Peubey, C., de Rosnay, P., Tavolato, C., Thépaut, J.-N., and Vitart, F. (2011). The era-interim reanalysis: configuration and performance of the data assimilation system. *Quarterly Journal of the Royal Meteorological Society*, 137(656):553–597.

Ganzeveld, L. and Lelieveld, J. (1995). Dry deposition parameterization in a chemistry general circulation model and its influence on the distribution of reactive trace gases. *J. Geophy. Res.*, 100:20999–21012.

Massman, W. J. (2004). Toward an ozone standard to protect vegetation based on effective dose: a review of deposition resistances and a possible metric. *Atmospheric Environment*, 38:2323–2337.

Nemitz, E., Sutton, M. A., Schjoerring, J. K., Husted, S., and Paul, W. G. (2000). Resistance modelling of ammonia exchange over oilseed rape. *Agricultural and Forest Meteorology*, 105:405–425.

Pilegaard, K., Skiba, U., Ambus, P., Beier, C., Brüggemann, N., Butterbach-Bahl, K., Dick, J., Dorsey, J., Duyzer, J., Gallagher, M., Gasche, R., Horvath, L., Kitzler, B., Leip, A., Pihlatie, M. K., Rosenkranz, P., Seufert, G., Vesala, T., Westrate, H., and Zechmeister-Boltenstern, S. (2006). Factors controlling regional differences in forest soil emission of nitrogen oxides (NO and N$_2$O). *Biogeosciences*, 3(4):651–661.

Rannik, U., Mammarella, I., Keronen, P., and Vesala, T. (2009). Vertical advection and nocturnal deposition of ozone over a boreal pine forest. *Atmospheric Chemistry and Physics*, 9(6):2089–2095.

Sogachev, A. (2009). A note on two-equation closure modelling of canopy flow. *Boundary-Layer Meteorol.*, 130:423–435.

Sogachev, A., Menzhulin, G., Heimannn, M., and Lloyd, J. (2002). A simple three dimensional canopy – planetary boundary layer simulation model for scalar concentrations and fluxes. *Tellus*, 54B:784–819.

---

## Author Response (AR2)

**Reply to comments on "Simulating ozone dry deposition at a boreal forest with a multi-layer canopy deposition model"**

December 27, 2016

We thank the reviewer's help on further improving our manuscript. Our reply for all the comments are shown below.

1. **Comments: I have indicated in my previous reviews that the manuscript requires linguistic revision. It still does.**

   We modified some texts in the manuscript to make it more clear and coherent. All the modifications are marked in the revised manuscript.

2. **Comments: When discussing deposition processes in the introduction, it would be useful to indicate that in many cases the results in question are measured at the same site, i.e. Hyytiälä, as studied in the present paper. Hyytiälä is mentioned on page 3, but later in the introduction you refer to the site as SMEAR II.**

   We modified the sentence to make it coherent and stressed that the study of Rannik et al. (2012) was conducted at the same site as in our study:

   "A study by Rannik et al. (2012), who conducted a detailed analysis of a long-term $O_3$ deposition flux measurement at a boreal forest station in Hyytiälä, Finland, ..."

   changed to

   "A study by Rannik et al. (2012), who conducted a detailed analysis of a long-term $O_3$ deposition flux measurement at the same site as in this study (SMEAR II, a boreal forest station in Hyytiälä, Finland), ...

3. **Comments: Following my comment, the authors have revised Eq. 15. It now includes an A term, which is "a unit scale factor which is set to 1 m2 m-3 here". It is not clear what the physical meaning of this factor is and why it could have a value different from 1, as the text implies.**

   We introduced two conductances to make the equations more clear:

   $$V_{d,i} = \text{LAI}_i V_{dveg,i} + \delta_{i1} V_{dsoil} \tag{1}$$

   $$V_{dveg,i} = \frac{1}{r_{veg,i}} \tag{2}$$

   $$V_{dsoil} = \frac{1}{r_{ac} + r_{bs} + r_{soil}}. \tag{3}$$

   $V_{dveg,i}$ is the layer-specific leaf surface conductance and $V_{dsoil}$ is the soil conductance. So the original Eq. 15 (evolution of $O_3$ concentration) changed to:

   $$\frac{\partial [O_3]}{\partial t} = \frac{\partial}{\partial z}\left(K_t \frac{\partial [O_3]}{\partial z}\right) - (V_{dveg}\text{LAD} + V_{dsoil}A_s)[O_3] + Q_{chem}. \tag{4}$$

$A_s$ (m$^2$ m$^{-3}$) is the soil area index which is the ratio between soil area and the model grid volume, hence it is non-zero only at the bottom layer which includes the soil surface.

4. **Comments: The discussion of the poor performance in the NL conditions should be improved. You refer to the possible role of vertical advection (P15) but do not explain why this would be important in these particular conditions. Please see you response (43) to my previous comments, which provides a much clearer discussion referring to the major role of soil deposition. The corresponding conclusion in the summary should be revised accordingly (P22/L19-20).**

We improved the explanation for the poor performance in the NL conditions:

"Rannik et al. (2009) revealed that the nighttime O$_3$ turbulent flux was affected by vertical advection of O$_3$. Therefore, when wet skin uptake is small for the condition NL, the vertical advection, which is not considered in the current model, could play a more crucial role in O$_3$ turbulent flux than deposition."

changed to

"Usually at nighttime RH is larger than 70% (Fig. 2), under this condition (NH condition), the wet skin uptake contributes more than 50% (Table 3) to the deposition flux. Therefore, the turbulent mixing above the ground which affects the deposition flux onto soil only plays a minor role on the deposition flux above the canopy. However, in NL condition which does not happen frequently, nearly all the deposition inside the canopy is caused by soil deposition. Hence, the difficulty of simulating the exchange processes near the surface may cause more uncertainty of simulating the deposition flux onto soil surface under NL condition than NH condition. Moreover, the vertical advection of O$_3$ could also affect the turbulent flux at nighttime (Rannik et al., 2009), which complicates the analysis."

We also modified the text in the summary:

"The possible reasons could be the limited data amount implying larger random uncertainty."

changed to

"The main reason could be the uncertainty of simulating the exchange processes near the ground in weak turbulent condition at nighttime when the soil deposition dominated the deposition flux inside the canopy."

5. **Comments: P20/L26: A typo in the unit. The explanation is unclear and does not help the reader.**

[revised manuscript text omitted]